# Learning to Reason via Mixture-of-Thought for Logical Reasoning

**Tong Zheng**[1,*] **Lichang Chen**[1,*], **Simeng Han**[2], **R. Thomas McCoy**[3], and **Heng Huang**[1]

[1]Department of Computer Science, University of Maryland, College Park, MD, USA
[2]Dept. of Computer Science, Yale University, New Haven, CT 06520
[3]Dept. of Linguistics, Yale University, New Haven, CT 06520

## Abstract

Human beings naturally utilize multiple reasoning modalities to learn and solve logical problems, *i.e.*, different representational formats such as natural language, code, and symbolic logic. In contrast, most existing LLM-based approaches operate with a single reasoning modality during training, typically natural language. Although some methods explored modality selection or augmentation at inference time, the training process remains modality-blind, limiting synergy among modalities. To fill in this gap, we propose *Mixture-of-Thought* (MoT), a framework that enables LLMs to reason across three complementary modalities: natural language, code, and a newly introduced symbolic modality, truth-table, which systematically enumerates logical cases and partially mitigates key failure modes in natural language reasoning. MoT adopts a two-phase design: (1) **self-evolving MoT training**, which jointly learns from filtered, self-generated rationales across modalities; and (2) **MoT inference**, which fully leverages the synergy of three modalities to produce better predictions. Experiments on logical reasoning benchmarks including FOLIO and ProofWriter demonstrate that our MoT framework consistently and significantly outperforms strong LLM baselines with single-modality chain-of-thought approaches, achieving up to +11.7pp average accuracy gain. Further analyses show that our MoT framework benefits both training and inference stages; that it is particularly effective on harder logical reasoning problems; and that different modalities contribute complementary strengths, with truth-table reasoning helping to overcome key bottlenecks in natural language inference.

## 1 Introduction

Large language models (LLMs) have demonstrated remarkable progress in logical reasoning tasks, especially propelled by methods like Chain-of-Thought (CoT) prompting (Wei et al., 2022). However, these CoT approaches predominantly rely on single reasoning modality, *i.e.*, natural language, even when employing ensemble methods (Li et al., 2023; Wang et al., 2025; 2022; Brown et al., 2024; Snell et al., 2025; Liang et al., 2023). Here we refer to a *modality* as a distinct thought paradigm[1] (*e.g.* natural language, symbolic, or code), which differs in representation and inference process. On the other hand, neuro-symbolic methods (Pan et al., 2023; Olausson et al., 2023; Ryu et al., 2025) utilize LLMs as translators and delegate reasoning to external symbolic solvers. Recent work combines CoT with symbolic reasoning via either selecting a single modality per instance (Xiong et al., 2024) or augmenting one modality with the other—while keeping reasoning confined to symbolic (Xu et al., 2024a) or natural language (Liu et al., 2024). **These methods combine modalities only during inference and ignore the synergy of different modalities during training**, thus failing to fully exploit the complementary strengths of different reasoning modalities.

This limitation contrasts sharply with human cognition: Humans naturally employ multiple reasoning modalities, flexibly switching among natural language explanations, code-based procedural thinking,

---

*Co-first authors.

[1]We use the terms thought paradigm and reasoning modality interchangeably.

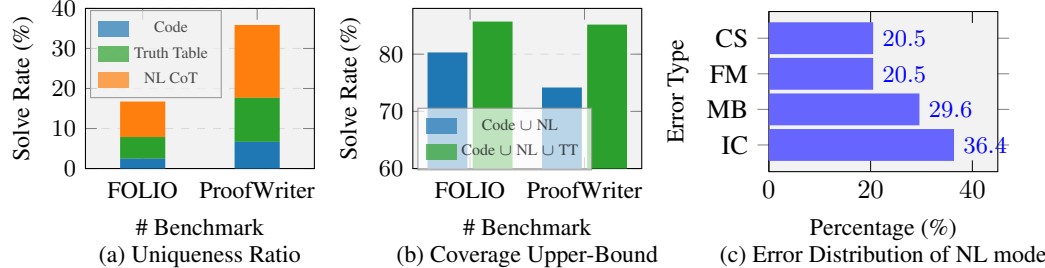

Figure 1: (a) Qwen-2.5-7B-Instruct solves $\simeq$20% of FOLIO and $\simeq$35% of ProofWriter exclusively per paradigm. (b) Code+NL+truth-table yields higher upper-bound coverage than code+NL alone (Xiong et al., 2024). (c) In NL modes, invalid-converse (IC) and missing-branch (MB) errors comprise $\simeq$66% of failures (CS: commonsense injection; FM: factual misquote). Percentages sum to more than 100% because some cases exhibit multiple error types. We provide illustrative examples in Appendix H.1

and formal symbolic manipulation, both when learning complex logical skills and when solving novel problems (Newell et al., 1972; Gentner, 1983; Larkin & Simon, 1987; Goldin, 1998). This cognitive versatility, the ability to represent and process information in diverse formats, is crucial for robust reasoning. Current LLMs, largely confined to single-modality training and inference, lack this flexibility. It raises a critical question: *Can LLMs achieve more robust and versatile logical reasoning by explicitly learning to operate across multiple complementary reasoning modalities?*

Addressing this question requires tackling two challenges: 1) It is still unclear which reasoning modalities should be included; the selected modalities must be complementary to make joint learning worthwhile. 2) Teaching an LLM with multiple modalities is non-trivial, as large aligned reasoning trajectories are scarce. Without those high-quality reasoning trajectories, we cannot teach LLMs to reason via multiple modalities. Our investigation reveals crucial insights for modality selection.

- **Natural language bottleneck.** Figure 1 (c) shows that nearly two thirds of CoT errors arise from *missing branches* and *invalid converse*, *i.e.*, poor exhaustive enumeration and complex inference (See examples in Appendix H.1). Truth-table reasoning, which systematically lists all possibilities, naturally complements this weakness; therefore, we incorporate a symbolic truth-table paradigm.
- **Code–NL complementarity.** Inspired by HybridMind (Yue et al., 2024; Xiong et al., 2024), where they show preliminary results that a code paradigm could complement NL reasoning, we also incorporate code as one reasoning modality into our framework.
- **Paradigm overlap & uniqueness.** Figure 1 (a-b) shows that 35.8% of ProofWriter items and 16.7% of FOLIO items are solved by *exactly one* paradigm, while the union of three reasoning modalities covers up to 85% of all instances—evidence that combining NL, code, and truth-table reasoning is necessary, outperforming the simple combination of code and NL (Xiong et al., 2024).

Building on these insights, we propose **M**ixture-**o**f-**T**hought (MoT), a human-inspired framework that enables LLMs to reason via three complementary reasoning modalities: NL, code, and symbolic; an example is shown in table 1 to illustrate each modality. Notably, we introduce a new truth-table-based symbolic reasoning where LLMs ground propositional variables, construct a partial truth table by pruning assignments that violate any premise, and infer the final answer by checking the truth table. Our MoT consists of two parts. One part is training: we propose a self-evolving MoT training algorithm, which improves the model's reasoning ability in each modality through joint iterative optimization (Figure 2 (a)). Another part is inference, where we generate responses under each modality and leverage a voting mechanism to produce the final answer (Figure 2 (b)). This simple strategy allows the model to combine diverse perspectives and make robust predictions.

Empirically, we show that across three base models—Gemma-2-2B-IT, Gemma-2-9B-IT, and Qwen-2.5-7B-Instruct—our MoT consistently surpasses the CoT baseline on ProofWriter (Tafjord et al., 2021) and FOLIO (Han et al., 2024), with an average accuracy gain of up to **+11.7pp**. Notably, our 9B-parameter MoT matches the results of GPT-4 + Logic-LM on FOLIO. Additional analyses show that 1) MoT training outperforms single-thought training; 2) Mixture-of-Thought sampling yields a higher oracle upper bound than single-thought sampling under the same inference budget 3) The gains grow with problem difficulty: MoT helps most on depth-5 and other harder problems; and 4) A fine-grained error study reveals a key natural-language bottleneck, *i.e.*, missing branches and frequent invalid converse errors, while the truth-table paradigm help resolve some cases of exactly these types.

Table 1: Illustration of the three complementary reasoning modalities, *i.e.*, natural-language CoT, code-based reasoning, and truth-table reasoning. We provide the corresponding outputs of LLMs in section G.

---

**Premise:** Peter Parker is either a superhero or a civilian. The Hulk wakes up when he is angry. If he wakes up, he will break a bridge. If a destroyer breaks a bridge, Peter is not a civilian. Peter wears a uniform when he is a superhero. Thor is a god. A god is not a destroyer. Thor will break a bridge when he is happy. If Thor is happy, the Hulk is angry.
**Question:** If Thor is happy, does Peter Parker wear a uniform?     **Options:** (A) True (B) False (C) Uncertain

| Natural Language Reasoning | Code-Based Reasoning (Abstract) | Truth Table Reasoning |
|---|---|---|
| ```<nl_cot>```
Step 1: given premises "If Thor is happy, the Hulk is angry." and "The Hulk wakes up when he is angry.", we can know "If Thor is happy, then hulk wakes up."

...

Step 5: given premise "Peter Parker is either a superhero or a civilian." and derivation "If Thor is happy, then Peter Parker is not a civilian", we can know "If Thor is happy, then Peter Parker is a superhero."
Step 6: given premise "Peter Parker wears a uniform when he is a superhero." and derivation "If Thor is happy, then Peter Parker is a superhero.", we can know "If Thor is happy, then Peter Parker wears a uniform"
```<end_of_nl_cot>```
```<answer>```
Answer: (A)
```<end_of_answer>``` | ``````
```class Hulk:```
```    def __init__(self, angry): ...```
```    def wakes_up(self): ...```
```    def breaks_bridge(self): ...```
```class Thor:```
```    def __init__(self, happy): ...```
```    def breaks_bridge(self): ...```
```class PeterParker:```
```    def __init__(self, is_superhero):```
```        ...```
```    def wears_uniform(self): ...```
```def apply_premises(thor, hulk, peter):```
```    ...```
```def run_inference(thor, hulk, peter):```
```    ...```
```def check_conclusion(...): ...```
```thor = Thor(happy=True)```
```hulk = Hulk(angry=False)```
```peter = PeterParker(...)```
```result = check_conclusion(...)```
```<end_of_code>```
```<answer>```Answer: (A)```<end_of_answer>``` | ```<truth_table>```
**Let:** $T$ = Thor happy, $H$ = Hulk angry, $A$ = wakes up, $B$ = bridge breaks, $C$ = Peter is civilian, $S$ = superhero, $U$ = wears uniform.

**Premises:**
$S \lor C, H \to A, A \to B, T \to H, T \to B,$
$B \to \neg C, S \to U.$

**Logical Chain (assume $T$ = True):**
$T \Rightarrow H \Rightarrow A \Rightarrow B$
$B \Rightarrow \neg C \Rightarrow C$ = False
$S \lor C \Rightarrow S$ = True $\Rightarrow U$ = True

**Truth Table:**

| $T$ | $H$ | $A$ | $B$ | $C$ | $S$ | $U$ |
\|---\|---\|---\|---\|---\|---\|---\|
\| True \| True \| True \| True \| False \| True \| True \|

```<end_of_truth_table>```
```<answer>```**Answer: (A)**```<end_of_answer>``` |

## 2   THE MIXTURE-OF-THOUGHT FRAMEWORK

In this section, we introduce 1) three complementary reasoning modalities for logical reasoning (Sec. 2.1); 2) our self-evolving training framework that jointly improves these reasoning modalities (Sec. 2.2); and 3) our mixture-of-thought inference strategy that combines diverse but complementary reasoning paths to derive robust final predictions (Sec. 2.3).

### 2.1   HUMAN-INSPIRED COMPLEMENTARY REASONING MODALITIES

Drawing inspiration from human cognition and error analysis in Figure 1, we argue that no single reasoning modality suffices for all logical challenges. Therefore, we equip a single model with three complementary modalities: natural language CoT, code CoT, and truth table CoT. Specifically, because natural-language CoT often misses branches or makes invalid-converse errors, we design a truth-table approach that explicitly enumerates truth assignments and thus complements these weaknesses. Table 1 illustrates how the three modalities solve a representative problem.

- **Natural Language CoT:** The model explains its reasoning in plain natural language, decomposing the problem into step-by-step justifications. This format is flexible and interpretable.
- **Code CoT:** The model first transforms a logical problem to a PYTHON code and then derives the answer based on the PYTHON code. We do not execute the code; instead, we treat it as a way to describe logic in a structured form.
- **Truth Table CoT:** The model first explicitly generates a truth table by defining predicates based on the premises and conclusion, then enumerating possible truth assignments, and finally checking which ones satisfy the conclusion.

These complementary modalities are jointly exploited in our self-evolving training (Sec. 2.2) and majority-vote inference (Sec. 2.3). We now detail the design of the Truth Table CoT approach.

**Truth-Table CoT: Challenges and Design.**   Two main challenges arise when enabling LLMs to reason with truth tables: 1) Exponential blow-up: the number of rows grows exponentially with the propositional variables, easily exceeding the context window and compute budget; 2) First-order grounding: practical tasks are given in first-order logic; one must ground variables, select a finite predicate set, and still ensure that the resulting (partial) truth table remains tractable. To address these challenges, we propose a two-step strategy: (i) grounding, which instantiates first-order formulas into a finite set of propositional predicates (Clarke et al., 2001; Wittocx et al., 2010), and (ii) reason to prune, which eliminates rows that violate any premise through reasoning via LLMs,

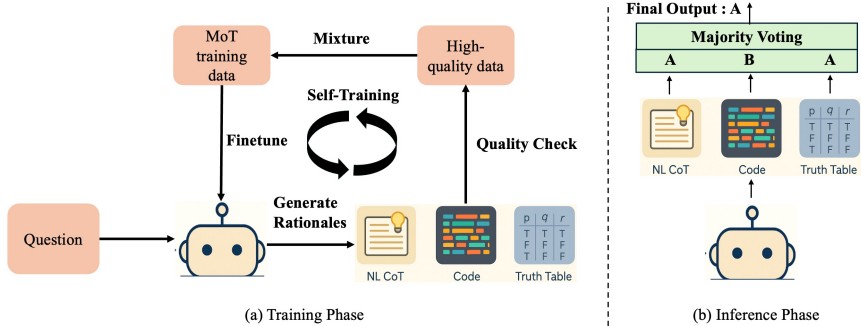

Figure 2: Illustration of our MoT Framework. (a) **Training phase** with three key steps: 1) **Rationale Generation** where given an initial seed dataset, LLM generates rationales across reasoning modalities (NL, Code, and Truth Table); 2) **Quality Checking and Merging** where generated rationales are checked for correctness and format consistency, then merged into high-quality MoT training data; and 3) **Finetuning** where the model is trained using the MoT data. These steps iteratively repeats, forming a self-evolving training cycle. (b) **Inference phase:** the trained model generates outputs for each reasoning modality and applies majority voting to yield the final prediction (e.g., A).

keeping partial truth table (see Table 1 and Appendix G.3). Finally, the LLMs derive the final output with the following rule: True if every surviving assignment satisfies the conclusion, False if none do, and Uncertain otherwise. Moreover, we assign modality-specific tags (e.g., `` ... `<end_of_code>`) to explicitly indicate the format during training and inference. The prompts are detailed in Appendix F.

## 2.2 SELF-EVOLVING MIXTURE-OF-THOUGHT TRAINING

Explicitly learning to reason across multiple complementary modalities, such as natural language, code, and symbolic truth tables, is non-trivial. A key challenge lies in the lack of annotated reasoning trajectories for each modality, especially for our newly introduced truth-table approach. Collecting labeled CoT traces for all of these modalities is also costly and often infeasible. To address this, we propose a self-evolving MoT training approach, which enables the model to operate across multiple complementary reasoning modalities by iteratively learning from its own generated reasoning traces.

Given the policy $M$, our goal is to maximize the following objective across the problems $x$ and modalities $\mathcal{T} \in \{\text{NL}, \text{Code}, \text{TruthTable}\}$ :

$$\mathbb{E}_{(x_i, y_i) \sim \mathcal{D}, \, t \sim \mathcal{T}, \, (z_i^t, \hat{y}_i^t) \sim M(\cdot | x_i, t, \mathcal{E}_t)} \left[ R(z_i^t, \hat{y}_i^t, y_i; t) \right], \tag{1}$$

where $\mathcal{D} = \{(x_i, y_i)\}_{i=1}^{D}$ is the dataset with problem $x_i$ and corresponding ground-truth $y_i$; $z_i^t$ and $\hat{y}_i^t$ be model-generated reasoning trace/answer with modality $t$ for $i$-th problem. To elicit the reasoning modality $t$, we design a small few-shot example set $\mathcal{E}_t$ for each $t$, and prepend the exemplar from the set to each problem $x_i$. Conditioned on $(x_i, t, \mathcal{E}_t)$, $z_i$ is sampled from policy $M$, followed by the prediction of the final answer $\hat{y}_i$. $R$ is the reward function and the design is detailed in the following.

**Reward Function $R$.** In preliminary experiments, we observe mismatch between tags and reasoning traces. This error leads to performance degradation, as different modalities negatively interfere with each other. Notably, this error is especially prevalent in the code modality. We define the reward as:

$$R(z, \hat{y}, y; t) = \begin{cases} 1, & y = \hat{y} \ \wedge \ \text{isValid}(z, t), \\ 0, & \text{otherwise}, \end{cases} \tag{2}$$

where the isValid function checks the format consistency by two standards: a) each trace should correctly include its modality's structural tag (e.g., `<end_of_nl_cot>` for nl) and b) for code traces, ensuring the presence of both a valid function definition (`def`) and a class definition (`class`); Following (Zelikman et al., 2022), we also filter out the traces with incorrect answer. We do not perform step-level verification of intermediate reasoning, as this would require additional external tools, e.g., LLMs judge (Zheng et al., 2023) and greatly slow down training. Instead, we find that simple checks (e.g., modality tags, basic code structures) already yield significant performance gains.

**Training.** We conduct multiple rounds of self-evolving training until performance saturates. $M_n$ is used to denote the policy in the $n$-th round with trainable parameters $\theta_n$. Leveraging the policy-gradient (Sutton et al., 1999) trick, we can easily obtain the gradient of eq. (1) as

$$\nabla J = \mathbb{E}_{\substack{(x_i,y_i)\sim\mathcal{D},t\sim\mathcal{T} \\ (z_i^t,\hat{y}_i^t)\sim M_{n-1}(\cdot|x_i,t,\mathcal{E}_t)}} \left[ R(z_i^t,\hat{y}_i^t,y_i;t)\, \nabla_{\theta_{n-1}} \log M_{n-1}(z_i^t,\hat{y}_i^t \mid x_i,t,\mathcal{E}_t) \right]. \quad (3)$$

In our current setting, the reward is binary (Eq. 2). This reduces the framework to a degenerate case of RL that can be equivalently seen as *rejection sampling plus supervised finetuning*, closely related to self-evolving or STaR training (Zelikman et al., 2022). Algorithm 1 and Figure 2 illustrate our multi-round training procedure. At round $n$, we prompt the model $M_{n-1}$ to generate a reasoning trace $z_i^t$ and a predicted answer $\hat{y}_i^t$ for each $x_i$ across all reasoning modalities $t \in \mathcal{T}$ (Line 4-9). It is worth noting that we use few-shot prompting only in the first round (Line 7); once the model has bootstrapped its own reasoning ability, all subsequent rounds run in zero-shot mode without additional exemplars (Line 9). We retain a sample only if it passes the quality filter (Line 11-13) and merge all surviving traces into $\mathcal{D}_{\text{all},n}^{\text{gen}}$ (Line 16). The updated model $M_n$, which is finetuned from $M_{n-1}$ on the filtered dataset $\mathcal{D}_{\text{all},n}^{\text{gen}}$ (Line 17). Unlike (Zelikman et al., 2022), which restarts from the base model each round, our training proceeds on-policy—learning from its own validated outputs. We demonstrate the effectiveness of this change in Appendix E.5.

---

**Algorithm 1** Self-Evolving MoT Training

**Input:** an LLM $M$; dataset $\mathcal{D} = \{(x_i,y_i)\}_{i=1}^{D}$; reasoning modality $\mathcal{T} = \{\text{NL}, \text{Code}, \text{TruthTable}\}$, Sampling times $S$, few-shot examples $\mathcal{E} = \{\mathcal{E}_{\text{NL}}, \mathcal{E}_{\text{Code}}, \mathcal{E}_{\text{TruthTable}}\}$
**Output:** Mixture-of-Thought enhanced model $M_N$

1: $M_0 \leftarrow M$
2: **for** $n = 1$ to $N$ **do**
3:     Initialize $\mathcal{D}_{\text{all},n}^{\text{gen}} \leftarrow \emptyset$; $\mathcal{D}_{\text{NL},n}^{\text{gen}} \leftarrow \emptyset$; $\mathcal{D}_{\text{Code,n}}^{\text{gen}} \leftarrow \emptyset$; $\mathcal{D}_{\text{TruthTable,n}}^{\text{gen}} \leftarrow \emptyset$
4:     **for all** $t \in \mathcal{T}$ **do**
5:         **for** $i = 1$ to $D$ **do**
6:             **if** n = 1 **then**
7:                 $z_i^t, \hat{y}_i^t \leftarrow M_{n-1}(x_i; t; \mathcal{E}_t; S)$
8:             **else**
9:                 $z_i^t, \hat{y}_i^t \leftarrow M_{n-1}(x_i; t; S)$
10:             **end if**
11:             **if** $R(z_i^t, \hat{y}_i^t, y_i; t)$=1 **then**
12:                 $\mathcal{D}_{t,n}^{\text{gen}} \leftarrow \mathcal{D}_{t,n}^{\text{gen}} \cup \{(x_i, z_i^t, y_i)\}$
13:             **end if**
14:         **end for**
15:     **end for**
16:     $\mathcal{D}_{\text{all,n}}^{\text{gen}} \leftarrow \text{Mix}(\mathcal{D}_{\text{NL,n}}^{\text{gen}}, \mathcal{D}_{\text{Code,n}}^{\text{gen}}, \mathcal{D}_{\text{TruthTable,n}}^{\text{gen}})$
17:     $M_n \leftarrow \text{Train}(M_{n-1}, \mathcal{D}_{\text{all,n}}^{\text{gen}})$
18: **end for**
19: **return** $M_N$

---

### 2.3 MIXTURE-OF-THOUGHT INFERENCE

To leverage the complementary strengths of three modalities, for each problem, we have three outputs corresponding to three modalities elicited by tagging, *i.e.*, `<nl_cot>`, ``, and `<truth_table>`, then we apply majority voting over outputs to decide the final answer. In case of ties, we randomly pick up the answer from one reasoning modality. We further explore the test-time scaling of MoT, and analyze its effectiveness in section 3.4.

## 3 EMPIRICAL EVALUATIONS

### 3.1 EXPERIMENTAL SETUPS

**Models.** To validate the effectiveness of our MoT, we select four widely-used LLMs across different sizes, reasoning abilities and model families: Qwen-2.5-7B-Instruct (Yang et al., 2024)/Qwen-3-4B-Instruct (Yang et al., 2025) and Gemma-2-2B-It/Gemma-2-9B-It (Team et al., 2024) as base models.

**Baselines.** Our approach is a kind of chain-of-thought approach. To this end, we select baselines from two folds: 1) neuro-symbolic approach and 2) chain-of-though approach. In the first category, we select Logic-LM (Pan et al., 2023) as a comparison. For the CoT approach, we select CoT (Wei et al., 2022) as a comparison. Since these approaches heavily rely on strong instruction-following capabilities, we directly cite their performance results from the original papers based on GPT-4.

**Dataset.** We select two logical reasoning benchmarks: ProofWriter (Tafjord et al., 2021) and FOLIO (Han et al., 2024) for evaluation. For ProofWriter, we select the hardest subset, which consists

Table 2: Accuracy (%) on the FOLIO and ProofWriter benchmarks. Our MoT training consistently improves the performance of each base model. Applying MoT inference further enhances performance across both benchmarks and all models. @3 denotes Self-Consistency approach (Wang et al., 2022) with three votes. **We provide full results of extra baselines (e.g., LoT) in Appendix E.3 & E.4 and robust analysis in Appendix E.1.**

| Model | Method Type | Reasoning Modality | FOLIO | ProofWriter | Avg |
|---|---|---|---|---|---|
| **(A) Prior SOTA Approach/Models** | | | | | |
| GPT-4 | *Logic-LM* | - | 78.9 | 79.7 | 79.3 |
| | *CoT (Vanilla)* | - | 70.6 | 68.1 | 69.4 |
| **(B) Base Model: Gemma-2-2B-It** | | | | | |
| Gemma-2-2B-It (3-Shot) | Single-Thought | Best (nl) | 42.4 | 39.8 | 41.1 |
| Gemma-2-2B-It @ 3 (3-Shot) | Single-Thought | Best (nl) | 45.3 | 38.8 | 42.1 |
| **MoT** (0-Shot) | Single-Thought | Best | 61.1 | 62.7 | 61.9 |
| **MoT** @ 3 (0-Shot) | Single-Thought | Best | 62.1 | 60.8 | 61.5 |
| **MoT** (0-Shot) | Mixture-of-Thought | All | **62.6** | **65.0** | **63.8** |
| **(C) Base Model: Gemma-2-9B-It** | | | | | |
| Gemma-2-9B-It (3-shot) | Single-Thought | Best (nl) | 69.5 | 61.2 | 65.4 |
| Gemma-2-9B-It @ 3 (3-shot) | Single-Thought | Best (nl) | 72.9 | 62.7 | 67.8 |
| **MoT** (0-shot) | Single-Thought | Best | 76.9 | 69.5 | 73.2 |
| **MoT** @ 3 (0-Shot) | Single-Thought | Best | 75.4 | 70.5 | 73.0 |
| **MoT** (0-shot) | Mixture-of-Thought | All | **78.9** | **70.7** | **74.8** |
| **(D) Base Model: Qwen2.5-7B-Instruct** | | | | | |
| Qwen2.5-7B-Instruct (3-shot) | Single-Thought | Best (nl) | 71.9 | 60.5 | 66.2 |
| Qwen2.5-7B-Instruct @ 3 (3-shot) | Single-Thought | Best (nl) | 73.4 | 65.8 | 69.6 |
| **MoT** (0-shot) | Single-Thought | Best | 75.9 | 69.2 | 72.6 |
| **MoT** @ 3 (0-shot) | Single-Thought | Best | 77.3 | 70.5 | 73.9 |
| **MoT** (0-shot) | Mixture-of-Thought | All | **78.3** | **71.8** | **75.1** |
| **(E) Base Model: Qwen3-4B-Instruct-2507** | | | | | |
| Qwen3-4B-Instruct-2507 (3-shot) | Single-Thought | Best (nl) | 83.3 | 79.7 | 81.5 |
| Qwen3-4B-Instruct-2507 @ 3 (3-shot) | Single-Thought | Best (nl) | 85.2 | 82.2 | 83.7 |
| **MoT** (0-shot) | Single-Thought | Best | 87.2 | 83.0 | 85.1 |
| **MoT** @ 3 (0-shot) | Single-Thought | Best | 87.2 | 83.8 | 85.5 |
| **MoT** (0-shot) | Mixture-of-Thought | All | **89.2** | **86.0** | **87.6** |

of 600 questions with reasoning depths of 5, the same as Pan et al. (2023). FOLIO is recognized for its high-quality export-made realistic test cases with more diverse reasoning depths ranging from 1-8. It consists of 203 questions. We utilize accuracy and pass@k as metrics.

**Training/Inference Details.** For each dataset, we collect 1000 training samples from the training set. We perform 2 or 3 rounds of self-evolving training. In each round, the model is fine-tuned for two epochs using a learning rate of 2e-5 and a batch size of 128. During the trajectory collection phase, the temperature, max_tokens, and sample count are set to 1.0, 2048, and 10, respectively. Notably, we use a high temperature (1.0) to encourage diverse trajectories. We sample each problem 10 times during trajectory collection to maximize coverage. Of all the generated traces, only the first single trajectory that satisfies our quality criteria is retained for the final training set. For evaluation, the temperature and max_tokens are configured to 0.7 and 2048, respectively. We do not perform hyperparameter tuning, so further optimization may yield even better performance. We run all experiments on 4 H100 GPUs. We employ vLLM engine (Kwon et al., 2023) to improve inference efficiency.

## 3.2 MAIN RESULTS

Table 2 displays the results on FOLIO and ProofWriter benchmarks. First, our Mixture-of-Thought (MoT) training with Single-Thought inference outperforms the corresponding base models by an average of **11.7pp** (from 41.1 to 61.9% for Gemma-2-2b-It, from 65.4% to 73.2% for Gemma-2-9b-It, from 66.2% to 72.6% for Qwen-2.5-7b-Instruct and from 81.5% to 85.1% for Qwen-3-4b-Instruct), demonstrating the effectiveness of our training strategy. When we further apply MoT inference, the MoT-trained model yields consistent additional gains of up to **2.1pp**. Notably, our 9B model achieves 78.9% accuracy on FOLIO, matching the performance of Logic-LM, which uses an external solver and focuses on close-sourced SoTA LLMs. Applying our MoT approach on reasoning model, e.g., Qwen3-4B-Instruct, achieves an average performance of 87.6%. Furthermore, MoT inference provides consistent gains over applying self-consistency to a single-thought setting, suggesting that the MoT mechanism offers complementary benefits. We provide a detailed performance of both base models and the corresponding MoT models, as well as stronger baselines, in Appendix E.3 & E.4.

Table 3: Accuracy (%) of different training strategies across reasoning modalities (Same Round). Shaded cells denote in-domain evaluation, i.e., testing on the same modalities during training. Avg. refers to the average performance using three modalities while Ensemble means the majority vote results on three modalities. Values underlined indicate that the model did not follow the instruction (*e.g.*, when asked to use Code, it still used NL).

| Training Approach | Param | Data | Code | NL_CoT | Truth Table | Avg. | Ensemble |
|---|---|---|---|---|---|---|---|
| *w/o Training* | | | | | | | |
| - | 9B | N/A | 56.7 | 69.5 | 63.6 | 63.3 | 66.0 |
| *Single-Thought Training* | | | | | | | |
| Single-Thought (Code) | 9B | - | 61.6 | 59.1 | 64.0 | 61.6 | 70.4 |
| Single-Thought (NL_CoT) | 9B | - | 52.7 | 73.9 | 69.0 | 65.2 | 73.4 |
| Single-Thought (Truth Table) | 9B | - | 53.2 | 69.0 | **71.9** | 64.7 | 71.9 |
| Single-Thought (Three Models Combined) | 3x9B | $\sim 3\times$ | 61.6 | 73.9 | **71.9** | 69.1 | 77.3 |
| *Mixture-of-Thought Training* | | | | | | | |
| Mixture-of-Thought (NL_CoT + Truth Table) | 9B | $\sim 2\times$ | 65.5 | 72.9 | 69.5 | 69.3 | 72.9 |
| Mixture-of-Thought (Truth Table + Code) | 9B | $\sim 2\times$ | 70.0 | 71.4 | 62.1 | 67.8 | 72.4 |
| Mixture-of-Thought (Code + NL_CoT) | 9B | $\sim 2\times$ | 70.9 | 70.0 | 74.4 | 71.8 | 74.9 |
| Mixture-of-Thought (Default, All) | 9B | $\sim 3\times$ | **73.9** | **76.9** | 70.0 | **73.6** | **78.9** |

## 3.3 MIXTURE-OF-THOUGHT TRAINING VS. SINGLE-THOUGHT TRAINING

In this section, we try to answer the key question: *Does MoT training truly offer benefits over Single-Thought training?* We have two baselines: 1) models trained on single-thought data and 2) models trained on partially MoT data, e.g., Code + NL. We evaluate both in-mode accuracy and cross-mode generalization. To enhance model's format following ability, we use 3-shot prompting to make model output the specific reasoning modality. Table 3 illustrates the results on FOLIO.

**SoT vs. MoT.** First, MoT training achieves the highest average accuracy across all three modalities, beating single-thought trained model, which indicates that our MoT training can jointly improve reasoning ability across all modalities. Second, MoT training can further push the performance boundary for each reasoning modality. For example, by using two of the three modalities, *i.e.*, Code and NL_CoT, the trained models outperform all single-thought baselines. This clearly indicates synergy between these three complementary modalities during training. Third, deploying one model for each modality is resource-expensive. In contrast, MoT training enables a single model to seamlessly switch among reasoning modalities based on prompts.

**Partial MoT vs. MoT.** Our default Mixture-of-Thought setting yields the best average performance and achieves the best accuracy by using two combined reasoning paradigms, which indicates that all the modalities are useful. This superiority is further reflected in the ensemble accuracy, where MoT achieves 78.9%. We provide more evidence in Sec. 4.3 and Appendix E.9.

**Additional Ablations for MoT Training.** We further give more analysis to show 1) robust and optimal design of the MoT framework (Appendix E.5); 2) MoT training is better than single-thought training with distillation data (Appendix E.6) and 3) MoT data outperform an equivalent amount of diverse single-thought CoT data (Appendix E.7). These results underscore the practical and broader value of our MoT framework.

## 3.4 TEST-TIME SCALING ACROSS REASONING MODALITIES

We investigate how different single-thought and MoT inference scale with an increased test-time budget. To do this, we first generate 128 responses from each model with each modality. Then we evaluate two sampling strategies: 1) *Single-Thought Sampling*: We randomly select $k$ responses from the 128 generated responses. and 2) *MoT Sampling*: Assuming there are $N_T$ reasoning modalities, we sample $\frac{k}{N_T}$ responses from each modality (so that the total number of responses is $k$). We choose $k$ ranging from 3 to 24 and have 10 runs for each setting.

**MoT framework vs. Single-thought Baseline.** We compare our Gemma-2-9b-It-MoT with Gemma-2-9b-It. Figure 3 (a) shows our MoT model with MoT sampling consistently outperforms Gemma-2-9b-It with single-thought sampling. When the sample budget is less than 20, the performance gap is significant. It suggests that our MoT approach significantly increases the response

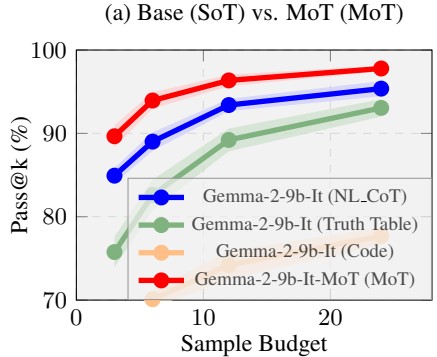
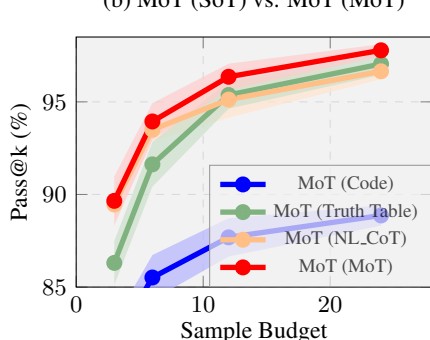

Figure 3: Pass@k vs. Sample Budget on FOLIO. (a) MoT-trained model with MoT sampling outperforms the base model (Gemma-2-9b-It) with SoT sampling. (b) Within the MoT-trained model, MoT sampling yields higher Pass@k than SoT sampling (NL_CoT, Truth Table, Code).

diversity, leading to a more efficient use of inference compute. We observe a consistent phenomenon in terms of averaged accuracy (Appendix E.8, Figure 6).

**Comparison of different modalities.** We further plot the scaling curves of our MoT model (based on Gemma-2-9B-It) under three reasoning modalities in Figure 3 (b). Here are insights: 1) While NL significantly outperforms the truth-table paradigm at low $k$, their theoretical upper bounds converge as $k$ increases; 2) The code paradigm exhibits the lowest upper bound among the three; 3) Across all values of $k$, our MoT framework consistently achieves the highest pass@k and attains the largest upper bound, indicating the largest potential of MoT trained models in test-time scaling.

## 4 FURTHER ANALYSIS

### 4.1 MIXTURE-OF-THOUGHT INFERENCE BENEFITS MORE FOR DIFFICULT PROBLEMS

We further identify the types of problems that benefit most from the proposed MoT inference approach. Specifically, we focus on problem difficulty, which can be effectively measured by the depth of reasoning. We conduct analysis on FOLIO and ProverQA (Qi et al., 2025). Figure 4 shows the performance of our MoT model with different reasoning modalities across reasoning depths.

We can see that MoT inference benefits more in solving more difficult logical problems. Our MoT model with MoT inference achieves an accuracy of 73.0% on challenging logical tasks with reasoning depths ranging from 5 to 8, outperforming each modality by a significant margin, with an average improvement of 9% points. However, such performance gains turn into slight degradation when dealing with easy problems. Similar phenomenon can be observed in ProverQA (See Figure 7 ).

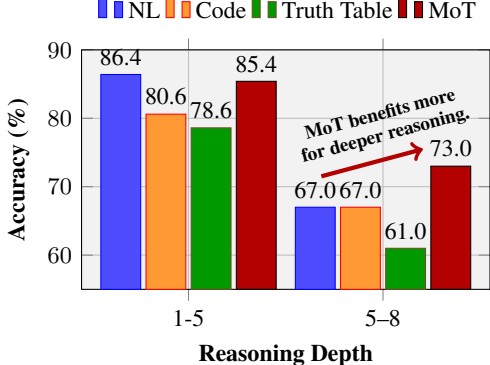

Figure 4: Performance comparison of different thought paradigms across reasoning depths. On FOLIO, MoT inference exhibits better performance on difficult problems.

### 4.2 ANALYSIS ON GENERALIZATION OF MOT

This section investigates a key question: *how does the Mixture-of-Thought (MoT) framework, primarily focused on improving logical reasoning tasks, generalize to non-logic reasoning tasks (OOD)?* Our evaluation focuses on two kinds of tasks: GSM8K and StrategyQA. Specifically, as our MoT mainly focuses on logical reasoning tasks, some of its reasoning modalities may be not suitable for these two tasks. To this end, we evaluate our MoT on these two tasks via NL reasoning modalities, ignoring other two reasoning modalities. The results are shown in Table 4. We can see the MoT framework exhibits robust generalization stability to non-logic OOD tasks when evaluated through the

Table 4: Performance comparison of Baseline vs. MoT (Single-Thought) across benchmarks.

| Model | Setting | Non-Logic (OOD) | | Logic (ID) | |
|---|---|---|---|---|---|
| | | StrategyQA | GSM8K | FOLIO | ProofWriter |
| **Gemma-2-2B-Instruct** | Baseline | 81.7 | 52.1 | 42.4 | 39.8 |
| | MoT (Single-Thought) | 81.2 | 51.5 | 61.1 | 62.7 |
| | Δ (MoT - Base) | -0.5 | -0.6 | +18.7 | +22.9 |
| **Gemma-2-9B-Instruct** | Baseline | 93.2 | 85.2 | 69.5 | 61.2 |
| | MoT (Single-Thought) | 92.3 | 84.2 | 76.9 | 69.5 |
| | Δ (MoT - Base) | -0.9 | -1.0 | +7.4 | +8.3 |
| **Qwen-2.5-7B-Instruct** | Baseline | 88.1 | 87.6 | 71.9 | 60.5 |
| | MoT (Single-Thought) | 89.5 | 86.9 | 75.9 | 69.2 |
| | Δ (MoT - Base) | +1.4 | -0.7 | +4.0 | +8.7 |

NL reasoning modality. This is evidenced by the fact that the performance changes (Δ) are minimal, predominantly remaining within ±1.4 pp. This demonstrates that MoT can significantly improve logical reasoning ability without compromising its general NL reasoning capabilities, successfully mitigating the risk of catastrophic forgetting.

**Discussions.** While we have shown that MoT achieves robust stability with minimal performance degradation, we also want to highlight that MoT is a general framework, which integrates natural language, code, and symbolic modalities into both training and inference. To directly enable the MoT framework to work on non-logic reasoning tasks, a promising way is to define a suitable instance of each modality for the specific domain. For example, for mathematical reasoning tasks like GSM8K, the symbolic modality can be instantiated as a equation derivations, algebraic simplification steps, enabling precise numerical manipulation.

### 4.3 Complementary, Uniqueness and Error Analysis

In this section, we quantify the complementary and uniqueness of our reasoning modalities and the training dynamics of our self-evolving MoT training. We focus on three metrics: 1) *Unique coverage*, *i.e.*, examples solved by exactly one modality; 2) *Complementarity coverage*, *i.e.*, examples solved by at least two modalities; and 3) *Oracle upper bound*, *i.e.*, examples solved by at least one modality.

Figure 1(a),(b) shows each modality's solve rate and oracle upper bound on ProofWriter and FOLIO. We further give a detailed unique and complementarity coverage and oracle upper bound in Table 11 in the Appendix. First, although our approach slightly reduces unique coverage compared to the baseline, both methods still achieve strong performance in this metric. Second, in terms of complementarity, our method increases the number of examples solved by multiple modalities—particularly on ProofWriter—demonstrating enhanced synergy. Third, by incorporating the truth-table paradigm alongside Code and NL, our model attains a higher oracle upper bound than prior work using only Code+NL, underscoring the benefit and necessity of the truth-table paradigm.

**Bottleneck of NL reasoning modality.** We perform a human evaluation of model outputs generated by natural language reasoning on the FOLIO dataset. We identify two major error patterns in the incorrectly solved cases: 1) failure to consider multiple cases when handling disjunction operations, such as "either/or"; 2) failure to utilize the transposition inference rule. For example, given A → B, the model might sometimes incorrectly produce ¬A → ¬B. Motivated by these observations and error types identified in prior work (Han et al., 2024; Olausson et al., 2023), we define four error categories: (i) invalid converse; (ii) missing branch; (iii) factual misquote; and (iv) incorporation of commonsense knowledge and design an automatic pipeline to assess model rationales. Figure 1(c) presents the results, showing that invalid converse and missing-branch errors together account for nearly 66% of all errors. These findings underscore the value of the Truth Table thought paradigm. We also provide error analysis after MoT training in Appendix E.2.

**Scenarios that Truth Table excels in.** We manually analyze all 13 examples (Table 11) that were solved only using the truth table paradigm and find that 1) 5 out of 13 problems require transposition; 2) 5 out of 13 problems contain disjunction or similar operations (e.g., 'Rock can fly, or is a bird, or cannot breathe') and 3) 2 out of 13 problems contain both. This indicates that Truth Table may indeed complement the NL paradigm to some extent. We give two examples in Appendix H.2.

**Scenarios that Code excels in.** We analyze the case that is solely solved by the code modality as shown in Table 11. We find the Code Modality is superior to the other two where high reasoning complexity is required. These examples are characterized by lots of premises and variables and require long inference chains. In this scenario, NL modality suffers from error accumulation or hallucination, while the TT modality faces challenges in building truth tables. The Code Modality fills this critical gap by providing a scalable and formally expressive encoding. However, we acknowledge that for complex problems that cannot be easily transformed into code structures, the Code Modality still falls short. This explains why code modality almost always performs worse when used alone.

## 5 RELATED WORK

**LLMs for Symbolic Reasoning.** Prior work has explored adapting LLMs to symbolic reasoning. One common approach treats LLMs as nl-to-fol translators, and then use an external symbolic prover to derive the final answer (Pan et al., 2023; Olausson et al., 2023; Callewaert et al., 2025; Ryu et al., 2025). While effective, this pipeline largely bypasses the model's internal reasoning capabilities, which our work seeks to fully leverage. To alleviate this problem, another line of work seeks to directly leverage LLMs' reasoning ability via CoT prompting (Wei et al., 2022) or structured search procedures such as Tree-of-Thought (Yao et al., 2023) and Cumulative Reasoning (Zhang et al., 2023), as well as improving determinacy via premise identification and prioritization (Sun et al., 2024). However, natural language remains inherently flexible and sometimes insufficient for structured reasoning. To bridge the gap between flexibility and formal rigor, recent work has explored combining natural and symbolic reasoning (Xu et al., 2024a; Liu et al., 2024; Xiong et al., 2024). These approaches often either rely on a primary reasoning modality (e.g., symbolic or NL), augmented with auxiliary signals from other representations (Xu et al., 2024a; Liu et al., 2024) or select one from multiple reasoning modalities (Xiong et al., 2024) at inference time. In contrast, our work 1) explicitly defines three kinds of reasoning paradigms covering natural language, symbolic and code-based reasoning. 2) goes beyond modality selection by jointly learning and inferring with all modalities, via a self-evolving MoT training and inference framework.

**Encouraging Diverse Thinking in Chain-of-Thoughts.** Previous work diversifies the CoT to further improve reasoning performance. A common strategy is to sample multiple outputs with higher temperatures (Wang et al., 2022; Brown et al., 2024), but this cannot guarantee true diversity (Wang et al., 2025). To address this, some work uses varied prompts—by task type (Wang et al., 2025), difficulty (Li et al., 2022), or strategy (Li et al., 2023; He et al., 2024)—and agent-based prompting via multi-agent debate (Liang et al., 2023; Hegazy, 2024) or self-reflection (Zhang et al., 2024) to elicit diverse CoTs. These methods diversify within one modality (NL or code). In contrast, we systematically introduce modality-level diversity—truth table, natural language, and code reasoning—which better aligns with the structural requirements of symbolic tasks and complements existing approaches. Recent work has also explored training smaller models on diverse CoTs generated by large LLMs (Ho et al., 2022; Puerto et al., 2024), though these approaches are limited to single-modality supervision and rely on external teacher models. In contrast, our method introduces modality-level diversity and requires no external supervision. We demonstrate that inter-modality diversity yields greater benefits for self-training than intra-modality diversity in Appendix E.7. Concurrent to our work, Chain-of-Reasoning (CoR) (Yu et al., 2025) also studies synergy of multiple reasoning modalities. However, CoR focuses on sequential synergy and targets mathematical reasoning. In contrast, our approach (i) focuses on logical reasoning and introduces TT reasoning modality, (ii) develops a self-evolving MoT training algorithm to bootstrap multi-modality capabilities, and (iii) exploits parallel synergy across modalities during both training and inference.

## 6 CONCLUSION

We presented **Mixture-of-Thought** (MoT), a unified framework for improving logical reasoning by enabling LLMs to reason through natural language, code-based, and symbolic (truth table) paradigms within a single system. Unlike previous work, our approach combines a self-evolving training process that fosters cross-paradigm synergy with an inference-time voting mechanism that aggregates complementary reasoning strategies. Extensive experiments on two challenging logical reasoning benchmarks, FOLIO and ProofWriter, demonstrate that MoT substantially outperforms strong baselines, particularly on complex, high-depth problems.

ACKNOWLEDGMENT

This work was partially supported by NSF IIS 2347592, 2348169, DBI 2405416, CCF 2348306, CNS 2347617, RISE 2536663.

ETHICS STATEMENT

Our Mixture-of-Thought (MoT) framework is designed to improve logical reasoning by integrating multiple complementary modalities (natural language, code, truth tables). We do not foresee any risks related to ethics issues.

REPRODUCIBILITY STATEMENT

We include detailed information about our approach in Section 2 and include experimental settings in Section 3 and Appendix D. We have give details in Section 3 and Appendix D to reproduce our results and we will open-source the code as soon as we collect our scripts for an easy way to reproduce our results.

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

## A  THE USE OF LLMs

We only use LLMs to polish the paper writing.

## B  LIMITATIONS AND FUTURE WORK

While our Mixture-of-Thought (MoT) framework demonstrates strong performance on logical reasoning tasks, we have not evaluated its effectiveness on other types of reasoning tasks, such as mathematical or commonsense reasoning. Additionally, our test-time scaling experiments suggest promising directions—such as dynamic mixture-of-thought sampling under budget constraints—but our current work still has not fully explored the benefits of complementary reasoning modalities. Further exploring these aspects could be important to further push the performance boundary of open-source models on reasoning.

We plan to further explore them in the two aspects:

- Extended to boarder tasks: currently our work cannot directly applied to reasoning tasks out of logical reasoning. This is because the reasoning modality we define in our work is specific for logical reasoning, *e.g.,* Truth Table. Therefore, we plan to define more general but complementary reasoning modality that can be applied to more general broader of reasoning tasks and further show how our MoT framework can further improve performance of reasoning tasks beyond logical reasoning.
- Adaptive parallel thinking via mixture-of-thoughts. An interesting question is *How can we fully leverage the benefits of complementary reasoning modalities during inference?* Recent work has shown that reinforcement learning can instill parallel thinking in LLMs (Zheng et al., 2025), enabling models to dynamically activate and coordinate multiple reasoning paths. This opens up a natural opportunity to integrate such adaptive parallel thinking with our MoT framework: treating each reasoning modality as an atomic unit, and allowing the model to dynamically trigger, combine, or prune modalities during inference. We believe that this unified view, blending MoT's modality-level diversity with dynamic parallelism, can lead to more flexible and efficient reasoning systems.

## C  BROADER RELATED WORK: SELF-EVOLVING TRAINING.

Self-evolving training techniques have been widely adopted to improve reasoning ability in LLMs, especially when there is lack of reasoning trajectories. Notably, Zelikman et al. (2022) propose a bootstrapping framework that iteratively generates and verifies reasoning trajectories based on the derived final answer, then fine-tunes the model on these self-labeled examples to improve reasoning performance with minimal human supervision. Following this idea, several works adapt self-evolving training to a wider range of tasks (Hosseini et al., 2024; Xiong et al., 2024; Zelikman et al., 2024b; Wang et al., 2024; Lin et al., 2024; Zelikman et al., 2024a; Poesia et al., 2024; Guan et al., 2025; Li et al., 2025). Additionally, researchers also explore improving the high-quality of rationales during STaR algorithm (Poesia et al., 2024; Guan et al., 2025; PENG et al., 2025), incorporating techniques such as formal verification, monte carlo tree search, and abstract-to-concrete prompting. While previous work primarily focuses on generating higher-quality reasoning paths within a single modality, our work explores a complementary direction: how to jointly evolve and coordinate reasoning across multiple thought paradigms.

## D  DETAILED EXPERIMENTAL SETTINGS

### D.1  DATASETS

In this work, we adopt three logical reasoning datasets: 1) FOLIO (Han et al., 2024), 2) ProofWriter (Tafjord et al., 2021), and 3) ProverQA (Qi et al., 2025), to evaluate the effectiveness of our MoT framework.

**FOLIO (Han et al., 2024).**  FOLIO provides both the training and validation subsets, consisting of 1003 and 203 samples, respectively. There are two subsets with different difficulties: 1) HybLogic:

contains 100 complex logical problems (5 – 8 reasoning steps) and 2) WikiLogic: contains 103 simper logical problems (1 – 5 reasoning steps). In this work, we sample 1000 training samples from the FOLIO training set as seed dataset for our self-evovling MoT training and evaluate both baselines and our trained model on the FOLIO validation set.

**ProofWriter (Tafjord et al., 2021).** ProofWriter is a synthetic dataset designed for evaluating the logical reasoning abilities of language models. It consists of multiple subsets, each containing logical reasoning problems of varying reasoning depths—from depth 0 (direct inference) up to depth 5 (requiring multi-step logical deductions). Following Pan et al. (2023), we select the most challenging subset (reasoning depth 5) to construct our training and test data. Specifically, we sample 1,000 instances from the training set provided by Pan et al. (2023) as our training data and adopt their original test set directly for fair evaluation.

**ProverQA (Qi et al., 2025).** ProverQA is a recently proposed logical reasoning benchmark, notable for its large scale, high quality, and diversity. It consists of three subsets, each corresponding to a different reasoning difficulty level (*i.e.*, reasoning depth). We select these subsets to evaluate the performance of our MoT framework across varying levels of reasoning complexity.

### D.2  TRAINING DETAILS

We conduct all experiments on 4 H100 GPUs with Alignment Handbook (Tunstall et al.). For each dataset, we sample 1,000 training examples and perform 2–3 rounds of self-evolving training. In each round, the model is fine-tuned for 2 epochs with a learning rate of 2e-5 and a batch size of 128. We do not perform hyperparameter tuning. Further tuning may lead to better performance. All experiments are run with a fixed seed, *i.e.,* 42, for reproducibility.

### D.3  INFERENCE DETAILS

We employ vLLM (Kwon et al., 2023) for efficient inference. During trajectory collection, we generate 10 reasoning traces per example using temperature 1.0, max_tokens 2048, and sampling count 10. To maximize coverage while ensuring quality, we retain only the first generated trace that passes our quality check. For evaluation, we set the temperature to 0.7 and max_tokens to 2048. All experiments are run with a fixed seed, *i.e.,* 42, for reproducibility.

### D.4  DETAILS OF "REASONING TO PRUNE"

The "Reasoning to Prune" step is the critical component in the Truth-Table CoT modality that resolves the computational inefficiency of enumerating the full $2^n$ truth table space. This process does not rely on iterative filtering; rather, it harnesses the LLM's few-shot reasoning capabilities to perform Constraint-Aware Deductive Reasoning, which implicitly prunes the logically inconsistent assignments.

**Implementation Through Few-Shot Prompting** We leverage few-shot prompting to guide the LLM into executing a specific, goal-oriented logical deduction. This approach compels the model to output only the assignments that constitute a logically consistent world, effectively generating the minimal required truth table row. The mechanism begins with Grounding and then operates via three integrated deductive processes:

1. **Grounding and Symbolic Mapping**: This initial phase involves the LLM performing symbolic mapping, instantiating the first-order problem's components into a finite set of propositional predicates. For example, the model establishes variables such as "Let: $T$ = Thor happy, $H$ = Hulk angry, $A$ = wakes up." This step formalizes the input for subsequent symbolic reasoning.

2. **Logical Chain Induction**: The LLM constructs a multi-step Logical Chain by identifying and utilizing the transitive properties present within the set of premises. For example, given premises $P_1 : A \Rightarrow B$ and $P_2 : B \Rightarrow C$, the LLM is prompted to perform the deduction $A \Rightarrow B \Rightarrow C$. This inductive process inherently establishes logical dependencies between the variables.

Table 5: Accuracy (%) on the FOLIO and ProofWriter benchmarks under three seeds (2025, 123 and 42). Our MoT training consistently improves the performance of each base model. Applying MoT inference further enhances performance across both benchmarks and all models. @3 denotes Self-Consistency approach (Wang et al., 2022) with three votes.

| Model | Method Type | Reasoning Modality | FOLIO | ProofWriter | Avg |
|---|---|---|---|---|---|
| **(A) Prior SOTA Approach** | | | | | |
| GPT-4 | *Logic-LM* | - | 78.9 | 79.7 | 79.3 |
| | *CoT (Vanilla)* | - | 70.6 | 68.1 | 69.4 |
| **(B) Base Model: Gemma-2-2B-It** | | | | | |
| Gemma-2-2B-It (3-Shot) | Single-Thought | Best (nl) | $41.9 \pm 1.1$ | $39.4 \pm 1.4$ | 40.7 |
| Gemma-2-2B-It @ 3 (3-Shot) | Single-Thought | Best (nl) | $42.7 \pm 2.5$ | $38.8 \pm 1.1$ | 40.8 |
| **MoT** (0-Shot) | Single-Thought | Best | $60.6 \pm 1.8$ | $61.4 \pm 1.8$ | 61.0 |
| **MoT** (0-Shot) | Mixture-of-Thought | All | $\mathbf{61.8 \pm 0.6}$ | $\mathbf{63.9 \pm 0.9}$ | **62.9** |
| **(C) Base Model: Gemma-2-9B-It** | | | | | |
| Gemma-2-9B-It (3-shot) | Single-Thought | Best (nl) | $70.5 \pm 1.1$ | $60.1 \pm 0.3$ | 65.3 |
| Gemma-2-9B-It @ 3 (3-shot) | Single-Thought | Best (nl) | $73.4 \pm 0.7$ | $62.5 \pm 0.2$ | 68.0 |
| **MoT** (0-shot) | Single-Thought | Best | $77.7 \pm 0.6$ | $68.4 \pm 0.8$ | 73.1 |
| **MoT** (0-shot) | Mixture-of-Thought | All | $\mathbf{79.3 \pm 0.4}$ | $\mathbf{71.0 \pm 0.5}$ | **75.2** |
| **(D) Base Model: Qwen2.5-7B-Instruct** | | | | | |
| Qwen2.5-7B-Instruct (3-shot) | Single-Thought | Best (nl) | $72.4 \pm 1.9$ | $60.8 \pm 1.9$ | 66.6 |
| Qwen2.5-7B-Instruct @ 3 (3-shot) | Single-Thought | Best (nl) | $73.1 \pm 0.5$ | $65.2 \pm 0.8$ | 69.2 |
| **MoT** (0-shot) | Single-Thought | Best | $76.5 \pm 0.8$ | $70.5 \pm 1.0$ | 73.5 |
| **MoT** (0-shot) | Mixture-of-Thought | All | $\mathbf{79.0 \pm 0.6}$ | $\mathbf{72.2 \pm 0.3}$ | **75.6** |

3. **Implicit Constraint Pruning**: The deduction is inherently constraint-aware. As the LLM sequentially derives the truth value for each variable (e.g., establishing $H = $ True), this assignment is immediately checked for consistency against all initial premises and all previously derived assignments. The LLM's goal is to derive and output the unique (or minimal) set of variable assignments that is logically sound with the entire premise set. By only outputting this consistent set, the process **implicitly prunes** all potential $2^n$ combinations that would have violated any premise, transforming the task from a search problem into a direct generation problem.

# E    ADDITIONAL EXPERIMENTAL RESULTS

## E.1    ROBUST ANALYSIS ON TABLE 2 WITH THREE SEEDS: 42, 123 AND 2025

Table 2 reports single-run results, which is the standard evaluation setting in prior logical reasoning works such as LINC (Olausson et al., 2023), Logic-LM (Pan et al., 2023), and LoT (Liu et al., 2024). To further access the robustness of our findings, we perform three runs under three different seeds, e.g., 42, 123 and 2025. We present the results in Table 5. We can observe consistent trend as that in Table 2. This confirms the reliability of our main claim.

## E.2    POST-MOT-TRAINING ERROR ANALYSIS

To complement the pre-training analysis in Figure 1(c), we conduct a post-MoT-training error-type analysis to examine whether MoT training also improves the NL modality itself. We evaluate four representative error categories: Invalid Converse (IC), Missing Branch (MB), Commonsense Injection (CI), and Factual Misquote (FC). Following the same annotation protocol as before, we compare the NL modality of the base model with the MoT-trained model.

Table 6: Post-MoT error-type counts for the NL modality.

| Model | IC | MB | CI | FC |
|---|---|---|---|---|
| Qwen-2.5-7B-Instruct | 22 | 18 | 12 | 12 |
| Qwen-2.5-7B-Instruct-MoT | 21 | 16 | 7 | 8 |

Overall, MoT training reduces all error categories, with notable improvements in CI and FC. However, IC and MB remain the most structurally challenging cases, indicating that NL alone cannot fully

resolve these errors. This highlights the importance of MoT-Inference, where NL, Code, and TT modalities collaboratively mitigate such difficult cases.

### E.3 Evaluating LLM Performance Across Reasoning Modalities on FOLIO and ProofWriter

Table 7: Performance of three models across reasoning modalities on FOLIO and ProofWriter.

| Model | FOLIO | | | ProofWriter | | |
|---|---|---|---|---|---|---|
| | NL | Code | Truth Table | NL | Code | Truth Table |
| Gemma-2-2B-It | 42.4 | 38.4 | 36.5 | 39.8 | 40.8 | 37.5 |
| + MoT training | 61.1 (18.7↑) | 61.1 (22.7↑) | 58.6 (22.1↑) | 62.7 (22.9↑) | 61.7 (20.9↑) | 60.2 (22.7↑) |
| Gemma-2-9B-It | 69.5 | 56.7 | 63.6 | 61.2 | 39.5 | 55.8 |
| + MoT training | 76.9 (7.4↑) | 73.9 (17.2↑) | 70.0 (6.4↑) | 68.5 (7.3↑) | 69.5 (30.0↑) | 66.7 (10.9↑) |
| Qwen-2.5-7B-Instruct | 71.9 | 62.1 | 69.0 | 60.5 | 42.3 | 53.0 |
| + MoT training | 75.9 (4.0↑) | 68.5 (6.4↑) | 71.9 (2.9↑) | 69.2 (8.7↑) | 66.7 (24.4↑) | 64.3 (11.3↑) |

Table 7 displays detailed results of baselines across reasoning modalities on FOLIO and ProofWriter. We can observe that LLMs owns uneven ability across these reasoning modalities. This also highlights the necessary of our self-evolving MoT training, which can equip LLMs with three complementary reasoning modalities. After self-evolving MoT training, all modalities show joint improvements. This effect is especially significant in smaller models, *i.e.*, Gemma-2-2B-It achieves up to a more than 20% increase in accuracy on average.

### E.4 Comparison with more Baselines on FOLIO and ProofWriter

Table 8: Comparison with more baselines on FOLIO and ProofWriter

| Method | Base Model | FOLIO (Acc %) | ProofWriter (Acc %) |
|---|---|---|---|
| HybridMind (Xiong et al., 2024) | GPT-3.5 | 76.6 | – |
| LINC (Olausson et al., 2023) | GPT-4 | 72.5 | – |
| Symbolic CoT (Xu et al., 2024b) | GPT-4 | 83.3 | 82.5 |
| Logic-of-Thoughts @ 5 | GPT-3.5 | 81.5 | 65.9 |
| Logic-of-Thoughts @ 5 | GPT-4 | 88.2 | 72.0 |
| MoT | Gemma-2-2b-It | 62.6 | 65.0 |
| MoT | Gemma-2-9b-It | 78.9 | 70.7 |
| MoT | Qwen2.5-7B-Instruct | 78.3 | 71.8 |

Table 8 presents a comparison between our approach and prior state-of-the-art systems. It demonstrates that our open-source MoT models nearly match the performance of leading closed-source prompting methods (*e.g.*, GPT-3.5 and GPT-4). This indicates that enabling LLMs to learn complementary reasoning modalities is a promising direction.

Table 9: Ablation studies on (1) policy strategy; and (2) mixing strategy.

| Ablation | Setting | FOLIO Accuracy (%) | | | |
|---|---|---|---|---|---|
| | | NL | Code | Truth Table | MoT |
| 1. Policy Strategy | Off-policy MoT | 55.2 | 54.7 | 53.7 | 56.7 |
| | On-policy MoT (default) | **61.1** | **61.1** | **58.6** | **62.6** |
| 2. Mixing Strategy | Random single-modality per question | 49.8 | 50.3 | 48.3 | 53.7 |
| | Direct mixing (default) | **61.1** | **61.1** | **58.6** | **62.6** |

### E.5 Ablation Studies

We perform ablation studies on three core components: 1) policy strategy, *i.e.,* on-policy vs. off-policy (Zelikman et al., 2022) and 2) mixing approach, *i.e.,* direct mixture vs. mixture by unique conclusion (randomly select single-modality per question).

Table 9 reports FOLIO accuracies under each setting. We make two key observations:

- **On-policy training yields consistent gains.** Switching from off-policy to on-policy increases single-modality CoT accuracy by approximately 5–6 pp (e.g., NL CoT from 55.2% to 61.1%) and raises MoT's final accuracy from 56.7% to 62.6%. This demonstrates the importance of updating the model with its most recent outputs.
- **Direct mixing outperforms random single-modality sampling.** Presenting all three modalities together boosts accuracy by 8–10 pp compared to randomly picking one modality per question (MoT: 62.6% vs. 53.7%). This indicates that joint exposure to multiple modalities provides stronger complementary signals than isolated examples.

### E.6 IMPACT OF QUALITY OF INITIAL TRAINING DATA: DISTILLATION + SINGLE-MODAL TRAINING VS. RAW DATA + MOT TRAINING

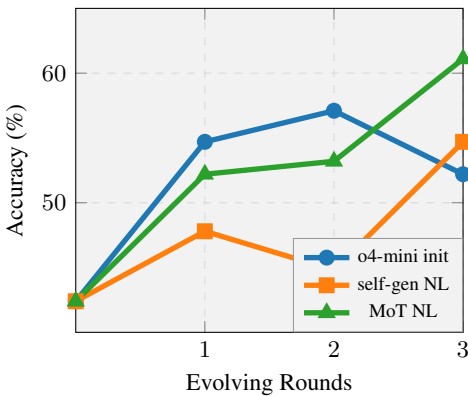

Figure 5: Accuracy (%) over three self-evolving rounds on the FOLIO benchmark for: distilled NL-CoT (first-round only), raw NL-CoT (no distillation), and MoT (no distillation). The performance is evaluated with NL-based reasoning.

Intuitively, the first-round data are crucial and have a strong impact on the efficacy of self-evolving training. Therefore, we are interested in the following question: *Can self-evolving single-thought training enhanced by first-round distillation outperform our self-evolving mixture-of-thought training without any distillation?* To answer this question, we compare the following settings: 1) Self-evolving single-thought (nl) training but with distillation data from o4-mini for first round training, which can provide a better initialization; 2) our MoT training without any distillation data; and 3) Self-evolving single-thought (nl) training without any distillation data. Figure 5 displays the results of Gemma-2-2b-It on FOLIO benchmark.

The key observations are: adding distillation data from stronger LLMs is beneficial for improving performance and convergence rate (blue line vs. orange line), but still lags behind our self-evolving MoT training (blue line vs. green line). This suggests the advantages of our self-evolving MoT training: 1) It requires no reliance on stronger—often more expensive—LLMs; 2) It provides a higher upper bound accuracy.

### E.7 FINETUNING WITH DIVERSE SINGLE-MODALITY COT VS. FINETUNING WITH MOT

Ho et al. (2022); Puerto et al. (2024) have explored that finetuning LLMs with diverse CoT can further improve the performance. A natural question then is: *given a fixed budget of training examples, which strategy yields better results? (1) self-training with $3N$ natural-language CoT samples, or (2) self-training with a total of $3N$ samples composed of $N$ examples from each of three modalities (NL, Code, Truth-Table)?*

We consider two settings to answer this question: 1) Self-evolving training with $3N$ natural-language CoT samples for *2 epochs per round* over *3 rounds*. We sample 10 reasoning traces per question with temperature of 1.0 and keep the 3 reasoning traces that satisfy our filtering criteria; 2) Self-evolving training with a total of $3N$ samples comprising $N$ examples from each of the three modalities (NL,

Code, Truth-Table) for *2 epochs per round* over *3 rounds*. We evaluate those trained model with natural language modality on FOLIO dataset.

Table 10: Accuracy (%) of Gemma-2-2b-It under three self-evolving regimes, with budgets of $N$ or $3N$ training samples. The accuracy is evaluated with NL-based reasoning on FOLIO benchmark. We can see self-evolving training with MoT achieves the best accuracy, demonstrating the benefit of modality-level diversity.

| # | Setting | Training Samples | Accuracy (%) |
|---|---------|------------------|--------------|
| 1 | NL_CoT | $N$ | 54.7 |
| 2 | NL_CoT | $3N$ | 57.1 |
| 3 | MoT data | $3N$ | 61.1 |

Table 10 shows the results. We can have the following observations: 1) finetuning with diverse NL CoT can indeed improve the performance (#1 vs. #2), which is consistent with findings from Ho et al. (2022); Puerto et al. (2024). 2) Finetuning with MoT data is more efficient than finetuning with same amount of diverse NL CoT data (#2 vs. #3). This indicates that the diversity of single-modality CoT data obtained by sampling with high temperature is not sufficient. By contrast, our MoT data, which leverages the complementarity of truth table, code and nl, can produce more diversity, and therefore improve the training efficiency.

### E.8 ADDITIONAL RESULTS ON TEST-TIME SCALING ACROSS REASONING MODALITIES

**MoT With Different Thought Paradigms**  Table 3 (b) illustrates the scaling behavior of our MoT model across different thought paradigms under varying sample budgets. We observe that code-based reasoning consistently lags behind all other paradigms, indicating its relatively poor performance and limited scalability.

Another interesting phenomenon is that natural language-based reasoning achieves relatively strong performance when the sample budget is small (e.g., $k < 20$), outperforming the truth table-based paradigm in this regime. However, as the sample budget increases (e.g., $k > 20$), truth table reasoning begins to match even outperform NL-based reasoning—highlighting its greater potential when more inference resources are available.

Notably, our MoT (ALL) approach offers a favorable trade-off between these two paradigms: it achieves strong performance under low-budget conditions, while delivering better performance when the sample budget is large.

**Accuracy vs. Sample Budget**  Figure 6 presents accuracy-vs-sample-budget curves across different reasoning paradigms. We find that our MoT (ALL) model—trained and inferred under the mixture-of-thought setting—consistently achieves the highest accuracy, outperforming all other approaches regardless of budget size. Additionally, our MoT model can benefit better from increased sample budget compared wiht all other approaches. Among individual paradigms, NL-CoT performs best under majority voting, while truth table reasoning is more stable but shows limited improvement with increased budget. Code-based reasoning remains the least effective. These results reinforce the value of our MoT framework.

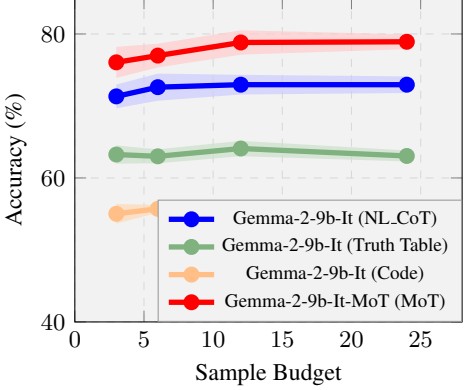

Figure 6: Accuracy vs. Sample Budget for different modes

### E.9 DETAILED COMPLEMENTARY, UNIQUENESS ANALYSIS

Across both ProofWriter and FOLIO benchmarks, our Mixture-of-Thought (MoT) model shifts away from single-paradigm reliance and toward multi-paradigm collaboration. First,

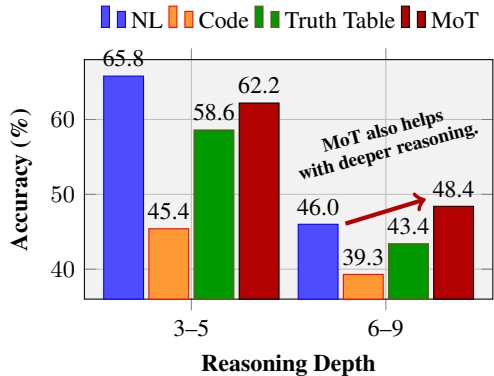

Figure 7: Performance comparison of different thought paradigms across reasoning depths. On ProverQA, MoT inference exhibits better performance on difficult problems.

the number of examples solved exclusively by the NL paradigm drops by over 50% (ProofWriter: from 109 to 55; FOLIO: from 18 to 8), and "Only TT correct" cases likewise decrease, indicating that MoT reduces brittle, single-mode reasoning. Second, pairwise overlaps (NL ∩ Code, NL ∩ TT, Code ∩ TT) all increase substantially—NL ∩ Code on ProofWriter rises by 76% (172 → 304), and similar gains appear on FOLIO—showing that MoT effectively combines different reasoning formats on the same instance. Finally, the overall coverage (Code ∪ NL ∪ TT) improves from 511 to 544 (+6.5%) on ProofWriter and from 174 to 181 (+4%) on FOLIO, demonstrating that MoT recovers difficult cases missed by the baseline. The consistent trends across two datasets confirm that encouraging multi-paradigm synergy yields more robust and comprehensive logical reasoning performance.

Table 11: Prediction Category Distribution on Two Benchmarks (Qwen-2.5-7B-Instruct vs Qwen-2.5-7B-Instruct-MoT).

| Category | ProofWriter | | FOLIO | |
|---|---|---|---|---|
| | Baseline | Our | Baseline | Our |
| *Single-paradigm only* | | | | |
| Only NL correct | 109 | 55 | 18 | 8 |
| Only Code correct | 40 | 32 | 5 | 6 |
| Only TT correct | 66 | 33 | 11 | 13 |
| *Pairwise overlap only* | | | | |
| NL ∩ Code only | 172 | 304 | 109 | 125 |
| NL ∩ TT only | 210 | 289 | 117 | 125 |
| Code ∩ TT only | 170 | 297 | 110 | 112 |
| Code ∪ NL | 445 | 511 | 163 | 168 |
| Code ∪ NL ∪ TT | 511 | **544** | 174 | **181** |

### E.10 PERFORMANCE VS. DIFFICULTY ON PROVERQA (QI ET AL., 2025)

Figure 7 displays the results.

## F FULL PROMPTS FOR MOT

The full prompts we utilized in this work are illustrated as follows:

---

**Full prompt used for Mixture-of-Thought**

```
You are a rigorous and logically precise AI assistant. Your
    task is to answer a logical reasoning problem strictly
    following one of three modes, as explicitly specified in
    the input. Only one mode will be present in the input.
    Follow that mode exclusively.

- Code Mode ( ... <end_of_code> <answer> ... <
    end_of_answer>)
  - If the input contains , translate the problem into
     Python code.
  - Execute the logic and derive the answer.

- Natural Language Chain-of-Thought Mode (<nl_cot> ... <
    end_of_nl_cot> <answer> ... <end_of_answer>)
```

```
      - If the input contains <nl_cot>, solve the problem step by
         step in natural language.

   - Truth Table Mode (<truth_table> ... <end_of_truth_table> <
      answer> ... <end_of_answer>)
      - If the input contains <truth_table>, construct a truth
         table and derive the answer from it.

   ### Rules
   - Only use the mode specified in the input. Do not switch modes
      .
   - Generate output strictly in the specified mode and format,
      with no additional text.
   - Enclose all reasoning strictly within the corresponding mode
      tags.
   - The final answer must be strictly enclosed in <answer> ... <
      end_of_answer>.
   - Do not provide any reasoning or explanations outside of the
      designated mode tags.

   The following is the problem you need to solve.

   <premises>
   {premises}
   </premises>

   <conclusion>
   {conclusion}
   </conclusion>

   <question>
   Is the following statement true, false, or uncertain? {
      conclusion}
   </question>

   <options>
   (A) True
   (B) False
   (C) Uncertain
   </options>

   <{tag}>
```

Full prompt used for Error Detection

```
"You must determine whether a rationale faithfully justifies
   the truth value of a conclusion given a set of premises.\n\
   n"
     "Faithful means all and only the steps actually used in
        deriving the conclusion:\n"
     "- are grounded in the given premises or prior derived
        steps,\n"
     "- apply valid inference rules (no illicit converse or
        contraposition),\n"
     "- cover every disjunction branch or quantifier case,\n"
     "- use no unstated assumptions, external knowledge, or
        background commonsense,\n"
     "- and correctly assess whether the conclusion is
        supported or contradicted by the premises.\n\n"
```

```
"You must also diagnose where and how the rationale fails
    when it is unfaithful, allowing trivial unused
    remarks to be overridden.\n\n"

"Error Types:\n"
"- Missing Branch: Failing to exhaustively consider all
    branches of a disjunction, conditionals, or
    quantified cases.\n"
"- Invalid Converse: Illicitly reversing the direction of
    a conditional (e.g., mistaking 'A →B' for 'B →A').\
    n"
"- Commonsense Injection: Using external background
    knowledge or commonsense not entailed or implied by
    the premises.\n"
"- Factual Misquote: Misrepresenting, distorting, or
    misquoting the explicit content of the premises.\n\n"

"Input (JSON):\n"
"{\n"
' "premises": "<string>",\n'
' "conclusion": "<string>",\n'
' "rationale": "<string>",\n'
' "label": "<string>",\n'
' "predict": "<string>"\n'
"}\n\n"

"Output (JSON):\n"
"{\n"
' "faithful": true | false,\n'
' "error_type": "<missing branch | invalid converse |
    commonsense injection | factual misquote>",\n'
' "error_location": "<e.g., Step 3, Clause 2>",\n'
' "override": true | false,\n'
' "analysis": "<brief summary explaining why the
    reasoning is faithful or unfaithful, citing specific
    logical failures>"\n'
"}\n\n"

"Notes:\n"
"- If multiple error types apply, list them all separated
    by commas.\n"
"- Always identify the first point in the rationale where
    the faithfulness failure occurs.\n"
"- Be concise, precise, and consistent in your labeling.\
    n\n"
"Input:\n"
```

# G   MORE EXAMPLES GENERATED BY OUR MOT

In this section, we show more examples generated by our MoT with different thought paradigms.

## G.1   NATURAL LANGUAGE BASED REASONING

Examples of NL-based Reasoning generated by our models.

```
The following is the problem you need to solve.

<premises>
```

```
All rabbits that can be spotted near the campus are cute.
Some turtles can be spotted near the campus.
The only animals that can be spotted near the campus are
    rabbits and squirrels.
If something is skittish, then it is not calm.
All the squirrels that can be spotted near the campus are
    skittish.
Rockie can be spotted near the campus, and it is calm.
</premises>

<conclusion>
Rockie is a turtle or cute.
</conclusion>

<question>
Is the following statement true, false, or uncertain? Rockie is
     a turtle or cute.
</question>

<options>
(A) True
(B) False
(C) Uncertain
</options>

<nl_cot>
Step 1: We know from the premise "The only animals that can be
    spotted near the campus are rabbits and squirrels." that
    Rockie, being spotted near the campus, must be either a
    rabbit or a squirrel.
Step 2: We also know from the premise "All rabbits that can be
    spotted near the campus are cute" that if Rockie is a
    rabbit, then Rockie is cute.
Step 3: The premise "All the squirrels that can be spotted near
     the campus are skittish" and "If something is skittish,
    then it is not calm" tell us that if Rockie is a squirrel,
    then Rockie is not calm.
Step 4: The premise "Rockie can be spotted near the campus, and
     it is calm" contradicts the conclusion from Step 3. This
    means Rockie cannot be a squirrel.
Step 5: Therefore, Rockie must be a rabbit.
Step 6: Since Rockie is a rabbit and all rabbits spotted near
    the campus are cute, we can conclude that Rockie is cute.

<end_of_nl_cot>
<answer>
The final answer is (A).
<end_of_answer>
```

## G.2 CODE BASED REASONING

Examples of Code-based Reasoning generated by our models.

```
The following is the problem you need to solve.

<premises>
Everyone at the mixer is a Grand Slam champion or an Oscar-
    nominated actor.
```

```
Every Grand Slam champion at the mixer is a professional tennis
    player.
All Oscar-nominated actors at the mixer are celebrities.
All professional tennis players at the mixer are athletes.
If a person at the mixer is a celebrity, then they are well
    paid.
If a person at the mixer is an athlete, then they are famous.
All well-paid people at the mixer live in tax havens.
Djokovic is at the mixer: if Djokovic is a famous athlete, then
    Djokovic is well-paid.
</premises>

<conclusion>
Djokovic does not live in a tax haven.
</conclusion>

<question>
Is the following statement true, false, or uncertain? Djokovic
    does not live in a tax haven.
</question>

<options>
(A) True
(B) False
(C) Uncertain
</options>


from typing import Optional

class Person:
    def __init__(self,
                 is_grand_slam_champion: Optional[bool] = None,
                 is_oscar_nominated_actor: Optional[bool] = None,
                 is_professional_tennis_player: Optional[bool] =
                     None,
                 is_celebrity: Optional[bool] = None,
                 is_athlete: Optional[bool] = None,
                 is_well_paid: Optional[bool] = None,
                 lives_in_tax_haven: Optional[bool] = None,
                 is_famous: Optional[bool] = None,
                 name: str = "Djokovic"):
        self.is_grand_slam_champion = is_grand_slam_champion
        self.is_oscar_nominated_actor = is_oscar_nominated_actor
        self.is_professional_tennis_player =
            is_professional_tennis_player
        self.is_celebrity = is_celebrity
        self.is_athlete = is_athlete
        self.is_well_paid = is_well_paid
        self.lives_in_tax_haven = lives_in_tax_haven
        self.is_famous = is_famous

def apply_premises(person: Person) -> bool:
    changed = False

    # Premise 1
    if person.is_grand_slam_champion is None and person.
        is_oscar_nominated_actor is None:
        person.is_grand_slam_champion = True
        changed = True

    # Premise 2
```

```
        if person.is_grand_slam_champion is True and person.
            is_professional_tennis_player is None:
            person.is_professional_tennis_player = True
            changed = True

        # Premise 3
        if person.is_oscar_nominated_actor is True and person.
            is_celebrity is None:
            person.is_celebrity = True
            changed = True

        # Premise 4
        if person.is_professional_tennis_player is True and person.
            is_athlete is None:
            person.is_athlete = True
            changed = True

        # Premise 5
        if person.is_celebrity is True and person.is_well_paid is
            None:
            person.is_well_paid = True
            changed = True

        # Premise 6
        if person.is_athlete is True and person.is_famous is None:
            person.is_famous = True
            changed = True

        # Premise 7
        if person.is_well_paid is True and person.lives_in_tax_haven
            is None:
            person.lives_in_tax_haven = True
            changed = True

        # Premise 8
        if person.name == "Djokovic" and person.is_famous is True
            and person.is_well_paid is None:
            person.is_well_paid = True
            changed = True

        return changed

    def run_inference(person: Person):
        while apply_premises(person):
            pass

    def check_conclusion(person: Person) -> str:
        run_inference(person)
        if person.name == "Djokovic" and person.lives_in_tax_haven
            is True:
            return "False"
        else:
            return "True"

    def func():
        person = Person(
            is_grand_slam_champion=None,
            is_oscar_nominated_actor=None,
            is_professional_tennis_player=None,
            is_celebrity=None,
            is_athlete=None,
            is_well_paid=None,
```

```
            lives_in_tax_haven=None,
            is_famous=None,
            name="Djokovic"
    )
    return check_conclusion(person)

if __name__ == '__main__':
    result = func()
    print("Conclusion: Djokovic does not live in a tax haven?",
        result)
<end_of_code>
<answer>
The final answer is (B).
<end_of_answer>
```

## G.3  TRUTH TABLE BASED REASONING

Examples of Truth Table-based Reasoning generated by our models.

```
The following is the problem you need to solve.

<premises>
People in this club who perform in school talent shows often
    attend and are very engaged with school events.
People in this club either perform in school talent shows often
    or are inactive and disinterested community members.
People in this club who chaperone high school dances are not
    students who attend the school.
All people in this club who are inactive and disinterested
    members of their community chaperone high school dances.
All young children and teenagers in this club who wish to
    further their academic careers and educational
    opportunities are students who attend the school.
Bonnie is in this club and she either both attends and is very
    engaged with school events and is a student who attends the
     school or is not someone who both attends and is very
    engaged with school events and is not a student who attends
     the school.
</premises>

<conclusion>
Bonnie performs in school talent shows often.
</conclusion>

<question>
Is the following statement true, false, or uncertain? Bonnie
    performs in school talent shows often.
</question>

<options>
(A) True
(B) False
(C) Uncertain
</options>

<truth_table>
Let:
    T: Bonnie performs in school talent shows often.
    E: Bonnie attends and is very engaged with school events.
```

```
   S: Bonnie is a student who attends the school.
   I: Bonnie is an inactive and disinterested community member.
   C: Bonnie chaperones high school dances.

Premises:
1. E →S
2. T ∨I
3. C →¬S
4. I →C
5. S →E
6. (E ∧S) ∨(¬E ∧¬S)

Logical chain (assuming T = False):
   From (2): T = False -> I = True
   From (4): I = True -> C = True
   From (3): C = True -> ¬S = True -> S = False
   From (6): S = False -> ¬E = True -> E = False
   From (1): E = False -> E→S holds vacuously
   -> No contradiction if T = False

Logical chain (assuming T = True):
   From (2): T = True -> I = False
   From (4): I = False -> C = False
   From (3): C = False -> ¬S = False -> S = True
   From (5): S = True -> E = True
   From (6): E = True ∧S = True -> (E∧S) = True
   -> No contradiction if T = True

Truth Table (remaining consistent rows):
| T | E | S | I | C |
|-------|-------|-------|-------|-------|
| True | True | True | False | False |
| False | False | False | True | True |

<end_of_truth_table>
<answer>
The final answer is (C).
<end_of_answer>
```

Examples of Truth Table-based Reasoning generated by our models.

```
The following is the problem you need to solve.

<premises>
All fir trees are evergreens.
Some objects of worship are fir trees.
</premises>

<conclusion>
Some evergreens are not objects of worship.
</conclusion>

<question>
Is the following statement true, false, or uncertain? Some
    evergreens are not objects of worship.
</question>

<options>
(A) True
(B) False
```

```
(C) Uncertain
</options>

<truth_table>
Let:
- F: An object is a fir tree.
- E: An object is an evergreen.
- W: An object is an object of worship.

Premises:
1. All fir trees are evergreens.
   F →E
2. Some objects of worship are fir trees.
   ∃x (W(x) ∧F(x))

Conclusion to Evaluate:
Some evergreens are not objects of worship.
∃x (E(x) ∧¬W(x))

Logical Chain:
1. From (1), F →E.
2. From (2), there exists some object a such that W(a) ∧F(a).
3. Since F(a) and F →E, it follows that E(a).
4. Therefore, we have E(a) and W(a).
5. We need ∃x (E(x) ∧¬W(x)).
6. The premises do not provide any information about whether
    there exists an evergreen that is not an object of worship.
7. Therefore, it is uncertain whether some evergreens are not
    objects of worship.

Truth Table (Key Possibilities):

| F | E | W | E ∧¬W |
| True | True | True | False |
| True | True | False| True |
| False| True | True | False |
| False| True | False| True |

Rows omitted: any with E= False (irrelevant to E∧¬W) or with F
    = True and E= False (violates F→E).

<end_of_truth_table>
<answer>
The final answer is (C).
<end_of_answer>
```

# H  ERROR ANALYSIS

## H.1  EXAMPLES OF ERROR TYPE

Example 1 of Missing Branches

```
<premises>
All employees who schedule a meeting with their customers will
    go to the company building today.
Everyone who has lunch in the company building schedules
    meetings with their customers.
```

```
Employees have lunch either in the company building or at home.
If an employee has lunch at home, they are working remotely
    from home.
All employees who are in other countries work remotely from
    home.
No managers work remotely from home.
James will appear in the company today if and only if he is a
    manager.
</premises>

<conclusion>
James has lunch in the company.
</conclusion>

<question>
Is the following statement true, false, or uncertain? James has
    lunch in the company.
</question>

<options>
(A) True
(B) False
(C) Uncertain
</options>

<nl_cot>
Step 1: From the premise \"All employees who schedule a meeting
    with their customers will go to the company building today
    ,\" we know that if an employee schedules a meeting with
    their customers, they will be in the company building today.

Step 2: From the premise \"Everyone who has lunch in the
    company building schedules meetings with their customers,\"
    we can infer that if an employee has lunch in the company
    building, they must schedule a meeting with their customers.

Step 3: From the premise \"Employees have lunch either in the
    company building or at home,\" we know that an employee can
    only choose one of these two options for lunch.

Step 4: From the premise \"If an employee has lunch at home,
    they are working remotely from home,\" we can deduce that
    having lunch at home means they are not in the company
    building and are working remotely.

Step 5: From the premise \"All employees who are in other
    countries work remotely from home,\" we know that employees
    in other countries do not have lunch in the company
    building.

Step 6: From the premise \"No managers work remotely from home
    ,\" we know that managers do not have lunch at home and
    thus do not work remotely.

Step 7: From the premise \"James will appear in the company
    today if and only if he is a manager,\" we can infer that
    if James is not a manager, he will not appear in the
    company today.
```

```
Step 8: Combining the information from steps 1-7, we can
     conclude that if James is a manager, he will be in the
     company today. Since James will only be in the company
     today if and only if he is a manager, and managers do not
     work remotely, James cannot be having lunch at home.
⇒Incorrect.  It only considers the case where James is a
     manger, missing another branch:  James is not a manger.

Step 9: Therefore, if James is not having lunch at home, he
     must be having lunch in the company building, and by step 2,
      he must be scheduling meetings with his customers, which
     means he will be in the company building today.

<end_of_nl_cot>
<answer>
The final answer is (A).
<end_of_answer>

<Label>
(C) Uncertain
</Label>
```

Example 2 of Both Missing Branches and Invalid Converse

```
<premises>
People in this club who perform in school talent shows often
     attend and are very engaged with school events.
People in this club either perform in school talent shows often
      or are inactive and disinterested community members.
People in this club who chaperone high school dances are not
     students who attend the school.
All people in this club who are inactive and disinterested
     members of their community chaperone high school dances.
All young children and teenagers in this club who wish to
     further their academic careers and educational
     opportunities are students who attend the school.
Bonnie is in this club and she either both attends and is very
     engaged with school events and is a student who attends the
      school or is not someone who both attends and is very
     engaged with school events and is not a student who attends
      the school.
</premises>

<conclusion>
Bonnie performs in school talent shows often.
</conclusion>

<question>
Is the following statement true, false, or uncertain? Bonnie
     performs in school talent shows often.
</question>

<options>
(A) True
(B) False
(C) Uncertain
</options>

<nl_cot>
```

```
Step 1: Analyze the given premises and the statement about
    Bonnie.
  - Premise 1: People in this club who perform in school
      talent shows often attend and are very engaged with
      school events.
  - Premise 2: People in this club either perform in school
      talent shows often or are inactive and disinterested
      community members.
  - Premise 3: People in this club who chaperone high school
      dances are not students who attend the school.
  - Premise 4: All people in this club who are inactive and
      disinterested members of their community chaperone high
      school dances.
  - Premise 5: All young children and teenagers in this club
      who wish to further their academic careers and
      educational opportunities are students who attend the
      school.
  - Bonnie's statement: Bonnie is in this club and she either
      both attends and is very engaged with school events and
      is a student who attends the school or is not someone
      who both attends and is very engaged with school events
      and is not a student who attends the school.
Step 2: Determine Bonnie's status in the club.
  - According to Bonnie's statement, she is either a very
      engaged and attending member or not such a member. This
      statement is a disjunction (or) of two conditions, which
      means Bonnie can only fit into one of the two
      categories described.
Step 3: Consider the implications for Bonnie based on her
    status in the club.
  - If Bonnie is a very engaged and attending member, she must
      perform in school talent shows often (Premise 1).
  ⇒Incorrect (Invalid Converse).  It incorrectly utilizes
      "Bonnie is a very engaged and attending member ->
      perform".
  - If Bonnie is not a very engaged and attending member, she
      must be an inactive and disinterested community member (
      Premise 2). Inactive and disinterested members must
      chaperone high school dances (Premise 4), which means
      they cannot be students who attend the school (Premise
      3).
  ⇒Incorrect (Missing Branches).  It fails to enumerate all
      possible cases stated in Premise.
Step 4: Evaluate the given statement.
  - The statement claims that Bonnie performs in school talent
      shows often.
  - Given the conditions and implications, if Bonnie is not a
      very engaged and attending member, she must be an
      inactive and disinterested member, which means she would
      be a chaperone and not a student, contradicting Premise
      5. Therefore, Bonnie must be a very engaged and
      attending member.
Step 5: Conclude based on the analysis.
  - Since Bonnie must be a very engaged and attending member,
      she performs in school talent shows often.
<end_of_nl_cot>
<answer>
The final answer is (A).
<end_of_answer>

<Label>
```

```
(C) Uncertain
</Label>
```

## Example 3 of Missing Branches

```
<premises>
All of Zaha Hadid's design styles that Max adores have
    interesting geometries.
No brutalist buildings that Max adores have interesting
    geometries.
Every style that Max adores is either Zaha Hadid's design style
     or Kelly Wearstler's design style.
All of Kelly Wearstler's design styles that Max adores are
    evocative.
All of Kelly Wearstler's design styles that Max adores are
    dreamy.
If a design by Max that he adores has interesting geometries,
    then the design is a brutalist building and evocative.
</premises>

<conclusion>
A design by Max is evocative and dreamy.
</conclusion>

<question>
Is the following statement true, false, or uncertain? A design
    by Max is evocative and dreamy.
</question>

<options>
(A) True
(B) False
(C) Uncertain
</options>

<nl_cot>
Step 1: Analyze the premise \"All of Zaha Hadid's design styles
     that Max adores have interesting geometries.\" This means
    if a design by Max adores is a Zaha Hadid design, then it
    has interesting geometries.

Step 2: Analyze the premise \"No brutalist buildings that Max
    adores have interesting geometries.\" This means if a
    design by Max adores is a brutalist building, it does not
    have interesting geometries.

Step 3: Analyze the premise \"Every style that Max adores is
    either Zaha Hadid's design style or Kelly Wearstler's
    design style.\" This means all designs by Max adores must
    be either Zaha Hadid's or Kelly Wearstler's.

Step 4: Analyze the premise \"All of Kelly Wearstler's design
    styles that Max adores are evocative.\" This means if a
    design by Max adores is a Kelly Wearstler design, then it
    is evocative.

Step 5: Analyze the premise \"All of Kelly Wearstler's design
    styles that Max adores are dreamy.\" This means if a design
```

```
     by Max adores is a Kelly Wearstler design, then it is
     dreamy.

Step 6: Analyze the premise \"If a design by Max that he adores
     has interesting geometries, then the design is a brutalist
     building and evocative.\" This means if a design by Max
     adores has interesting geometries, then it is both a
     brutalist building and evocative.

Step 7: From Step 1 and Step 6, if a design by Max adores is a
     Zaha Hadid design and has interesting geometries, it must
     also be a brutalist building and evocative.

Step 8: From Step 2, if a design by Max adores is a brutalist
     building, it does not have interesting geometries. So, a
     design by Max adores cannot be both a brutalist building
     and have interesting geometries.

Step 9: Since a design by Max adores cannot have both
     interesting geometries (which would make it a brutalist and
      evocative) and not have interesting geometries (which
     would make it not a brutalist), the design by Max adores
     cannot have interesting geometries.
```
⇒Incorrect (Missing Branches).  It fails to enumerate all
     possible cases stated in Premise 3.
```
Step 10: Since the design by Max adores does not have
     interesting geometries, according to Step 4 and Step 5, it
     cannot be a Kelly Wearstler design. Therefore, it must be a
      Zaha Hadid design.

Step 11: Since the design by Max adores is a Zaha Hadid design,
      it must have interesting geometries (from Step 1), but
     this contradicts the conclusion that it cannot have
     interesting geometries (from Step 8).

Step 12: Given the premises, it is impossible for a design by
     Max adores to be both evocative and dreamy, as it would
     require having interesting geometries and not having them
     at the same time.
<end_of_nl_cot>
<answer>
The final answer is (B).
<end_of_answer>

<Label>
(A) True
</Label>
```

## H.2 Examples uniquely solved by Truth Table

```
Examples uniquely solved by Truth Table

<premises>
"Stranger Things" is a popular Netflix show.
If a Netflix show is popular, Karen will binge-watch it.
```

```
If and only if Karen binge-watches a Netflix show, she will
    download it.
Karen does not download "Black Mirror."
"Black Mirror" is a Netflix show.
If Karen binge-watches a Netflix show, she will share it with
    Lisa.
</premises>

<conclusion>
"Black Mirror" is popular.
</conclusion>

<question>
Is the following statement true, false, or uncertain? "Black
    Mirror" is popular.
</question>

<options>
(A) True
(B) False
(C) Uncertain
</options>

<nl_cot>
Step 1: Given that \"Stranger Things\" is a popular Netflix
    show and if a Netflix show is popular, Karen will binge-
    watch it, we know that Karen would binge-watch \"Stranger
    Things\" if it were popular. ⇒Correct.

Step 2: However, the popularity of \"Stranger Things\" does not
     provide any direct information about the popularity of \"
    Black Mirror.\" ⇒Correct.

Step 3: We know that \"Black Mirror\" is a Netflix show, but we
     do not have any information that directly links its
    popularity to Karen's actions. ⇒Correct.

Step 4: The statement that if Karen binge-watches a Netflix
    show, she will download it, and the fact that Karen does
    not download \"Black Mirror,\" imply that Karen does not
    binge-watch \"Black Mirror.\" ⇒Correct.

Step 5: Since Karen does not binge-watch \"Black Mirror,\" and
    if a show is popular, Karen will binge-watch it, we cannot
    conclude that \"Black Mirror\" is popular based on Karen's
    actions. ⇒Incorrect.  Fail to utilize
    Transposition.

Step 6: The conclusion \"Black Mirror\" is popular cannot be
    definitively supported or refuted with the given premises.
    ⇒Incorrect.

<end_of_nl_cot>
<answer>
The final answer is (C). ⇒Incorrect.
<end_of_answer>"

<truth_table>
Let:
    P: \"Stranger Things\" is a popular Netflix show.
    Q: \"Black Mirror\" is a popular Netflix show.
    B: Karen will binge-watch a Netflix show.
```

```
    D: Karen will download a Netflix show.
    S: Karen will share a Netflix show with Lisa.
Premises:
    P = True (Given) ⇒Correct.
    If P, then B (Given) ⇒Correct.
    If B, then D (Given) ⇒Correct.
    If and only if B, then D (Given) ⇒Correct.
    \"Black Mirror\" is a Netflix show (Given) ⇒Correct.
    If B, then S (Given) ⇒Correct.
Conclusion to Evaluate:
    Q (Is \"Black Mirror\" popular?) ⇒Correct.
Logical Chain:
    From (5), \"Black Mirror\" is a Netflix show. ⇒Correct.
    From (2) and (3), if \"Black Mirror\" is popular (Q), then
        Karen will binge-watch it (B). ⇒Correct.
    From (4), if Karen binge-watches \"Black Mirror,\" then she
        will download it (D). ⇒Correct.
    However, from (6), we know that Karen does not download \"
        Black Mirror,\" so D = False. ⇒Correct.
    Since B implies D (from (3) and (4)), and D is False, B must
         also be False. ⇒Correct.
    Since B is False (Karen does not binge-watch \"Black Mirror
        \"), Q (popularity of \"Black Mirror\") cannot be True.
         ⇒Correct.
Truth Table (Key Possibilities):
| Q (Q: \"Black Mirror\" is popular) | B (B: Karen will binge-
    watch \"Black Mirror\") | D (D: Karen will download \"Black
     Mirror\") |
| False | False | False |
| True | True | True |
Given that D is False, Q must be False. ⇒Correct.
<end_of_truth_table>
<answer>
The final answer is (B). ⇒Correct.
<end_of_answer>


from typing import Optional

class Show:
    def __init__(self, name: str, is_popular: Optional[bool] =
        None, is_downloaded: Optional[bool] = None, is_binged:
         Optional[bool] = None):
        self.name = name
        self.is_popular = is_popular
        self.is_downloaded = is_downloaded
        self.is_binged = is_binged

def apply_premises(show: Show) -> bool:
    changed = False

    # Premise 1: "Stranger Things" is a popular Netflix show.
    if show.name == "Stranger Things": ⇒Correct.
        if show.is_popular is None:
            show.is_popular = True
            changed = True

    # Premise 2: If a Netflix show is popular, Karen will binge-
        watch it.
    if show.is_popular is True and show.name != "Black Mirror":
        ⇒Correct.
```

```
        if show.is_binged is None:
            show.is_binged = True
            changed = True

    # Premise 3: If and only if Karen binge-watches a Netflix
        show, she will download it.
    if show.is_binged is True: ⇒Correct.
        if show.is_downloaded is None:
            show.is_downloaded = True
            changed = True

    # Premise 4: "Black Mirror" is a Netflix show. ⇒Incorrect.
        1.  Do not encode Transposition.  2.  It directly set
        show.is_popular = False., so even if it set it correctly, as
        it do not include Transposition, code approach still
        cannot work correctly.
    if show.name == "Black Mirror":
        if show.is_popular is None:
            show.is_popular = False
            changed = True

    # Premise 5: If Karen binge-watches a Netflix show, she will
        share it with Lisa.
    if show.is_binged is True: ⇒Correct.
        if show.is_downloaded is True:
            if show.name == "Black Mirror":
                if show.is_popular is False:
                    show.is_popular = True
                    changed = True

    return changed

def run_inference(show: Show):
    while apply_premises(show):
        pass

def check_conclusion(show: Show) -> str:
    run_inference(show)
    if show.name == "Black Mirror":
        if show.is_popular is True:
            return "True"
        elif show.is_popular is False:
            return "False"
        else:
            return "Uncertain"
    else:
        return "Uncertain"

def func():
    show = Show(name="Black Mirror", is_popular=None,
        is_downloaded=None, is_binged=None)
    return check_conclusion(show)

if __name__ == '__main__':
    result = func()
    print("Conclusion: 'Black Mirror' is popular?", result)

<end_of_code>
<answer>
The final answer is (C). ⇒Incorrect.
<end_of_answer>
```

Examples uniquely solved by Truth Table

```
<premises>
All of Peter's pets that are mammals are also animals.
All monkeys are mammals.
Peter's pets are all either monkeys or birds.
Peter's birds can fly.
All animals that are Peter's pets can breathe.
If Peter's pet can fly, then it has wings.
Rock is Peter's pet.
Rock can fly, or Rock is a bird, or Rock cannot breathe.
</premises>

<conclusion>
Rock is a monkey.
</conclusion>

<question>
Is the following statement true, false, or uncertain? Rock is a
    monkey.
</question>

<options>
(A) True
(B) False
(C) Uncertain
</options>

<nl_cot>
Step 1: From the premise \"All of Peter's pets that are mammals
    are also animals,\" we know that any mammal pet of Peter's
    is an animal.

Step 2: From the premise \"All monkeys are mammals,\" we know
    that if something is a monkey, it is also a mammal.

Step 3: From the premise \"Peter's pets are all either monkeys
    or birds,\" we know that Rock, being a pet of Peter's, must
     be either a monkey or a bird.

Step 4: From the premise \"Peter's birds can fly,\" we know
    that if something is a bird, it can fly.

Step 5: From the premise \"All animals that are Peter's pets
    can breathe,\" we know that any pet of Peter's, whether a
    bird or a mammal, can breathe.

Step 6: From the premise \"If Peter's pet can fly, then it has
    wings,\" we know that if a pet of Peter's can fly, it must
    have wings.

Step 7: From the premise \"Rock can fly, or Rock is a bird, or
    Rock cannot breathe,\" we have a disjunction. We need to
    evaluate each part of this disjunction:\n- If Rock can fly,
     then it must have wings (Step 6), and it must be a bird (
    Step 4), because only birds can fly among Peter's pets.\n-
    If Rock is a bird, it can fly (Step 4), and it can breathe
    (Step 5).\n- If Rock cannot breathe, this contradicts the
    fact that all of Peter's pets can breathe (Step 5).
    ⇒InCorrect.  If Rock can fly, then it must have wings
    (Step 6), and it must be a bird (Step 4):  this step
```

```
        incorrectly utilizes rule fly -> birds to derive rock must
        be a bird.

    Step 8: Given the disjunction in Step 7, the only consistent
        option is that Rock is a bird, because if Rock can fly, it
        must be a bird, and if Rock cannot breathe, it contradicts
        the premises. Therefore, Rock is a bird.\n\nStep 9: Since
        Rock is a bird, and all of Peter's pets are either monkeys
        or birds (Step 3), Rock cannot be a monkey.

    <end_of_nl_cot>
    <answer>
    The final answer is (B).
    <end_of_answer>

    <truth_table>
    Let:
        M: Rock is a monkey
        B: Rock is a bird
        F: Rock can fly
        A: Rock is an animal
        W: Rock has wings
        R: Rock can breathe

    Premises:
        1. All of Peter's pets that are mammals are also animals. (
            Not directly about Rock)
        2. All monkeys are mammals. (Not directly about Rock)
        3. Peter's pets are all either monkeys or birds. (Not
            directly about Rock)
        4. Peter's birds can fly. (B →F)
        5. All animals that are Peter's pets can breathe. (A →R)
        6. If Peter's pet can fly, then it has wings. (F →W)
        7. Rock is Peter's pet.
        8. Rock can fly, or Rock is a bird, or Rock cannot breathe.
            (F ∨B ∨¬R)

    Conclusion to Evaluate:
        Rock is a monkey (M)
```

⇒ **Correct.** The model correctly produce two cases that lead to different conclusions, *i.e.,* True and False. Therefore, it finally produces Uncertain as the final answer.

```
    Truth Table (Key Possibilities):

    | M (Monkey) | B (Bird) | F (Can Fly) | A (Animal) | W (Wings)
        | R (Can Breathe) |
    | True | False | True | True | True | True |
    | False | True | True | True | True | True |
    <end_of_truth_table>
    <answer>
    The final answer is (C).
    </end_of_answer>

    
    from typing import Optional

    class Pet:
```

```
    def __init__(self,
             is_monkey: Optional[bool] = None,
             is_bird: Optional[bool] = None,
             can_fly: Optional[bool] = None,
             has_wings: Optional[bool] = None,
             is_animal: Optional[bool] = None):
        self.is_monkey = is_monkey
        self.is_bird = is_bird
        self.can_fly = can_fly
        self.has_wings = has_wings
        self.is_animal = is_animal

def apply_premises(pet: Pet) -> bool:
    changed = False

    # Premise 1: All of Peter's pets that are mammals are also
        animals.
    # Since all of Peter's pets are either monkeys or birds, and
         monkeys are mammals,
    # this premise does not directly affect the state of the pet
        .

    # Premise 2: All monkeys are mammals.
    if pet.is_monkey is True and pet.is_animal is not True:
        pet.is_animal = True
        changed = True

    # Premise 3: Peter's pets are all either monkeys or birds.
    if pet.is_monkey is None and pet.is_bird is None:
        pet.is_monkey = False
        pet.is_bird = False
        changed = True

    # Premise 4: Peter's birds can fly.
    if pet.is_bird is True and pet.can_fly is not True:
        pet.can_fly = True
        changed = True

    # Premise 5: All animals that are Peter's pets can breathe.
    if pet.is_animal is True and pet.can_fly is not True:
        pet.can_fly = True
        changed = True

    # Premise 6: If Peter's pet can fly, then it has wings.
    if pet.can_fly is True and pet.has_wings is not True:
        pet.has_wings = True
        changed = True

    # Premise 7: Rock is Peter's pet.
    # This does not change the state of the pet but indicates
        that the pet is Rock.

    # Premise 8: Rock can fly, or Rock is a bird, or Rock cannot
         breathe.
    if pet.name == "Rock":
        if pet.can_fly is None:
            pet.can_fly = True
            changed = True
        if pet.is_bird is None:
            pet.is_bird = True
            changed = True
        if pet.can_fly is True and pet.has_wings is not True:
```

```
            pet.has_wings = True
            changed = True
    ⇒ Incorrect.  The model struggles to correctly interpret
        premises that involve multiple possible branches of
        reasoning, such as disjunctions (∨).  It often commits
        prematurely to one or more branches instead of
        preserving logical uncertainty.

    return changed

def run_inference(pet: Pet):
    while apply_premises(pet):
        pass

def check_conclusion(pet: Pet) -> str:
    run_inference(pet)
    if pet.name == "Rock" and pet.is_monkey is True:
        return "True"
    else:
        return "False"

def func():
    pet = Pet(
        is_monkey=None,
        is_bird=None,
        can_fly=None,
        has_wings=None,
        is_animal=None
    )
    pet.name = "Rock"
    return check_conclusion(pet)

if __name__ == '__main__':
    result = func()
    print("Conclusion: Rock is a monkey?", result)
<end_of_code>
<answer>
The final answer is (B).
</end_of_answer>
```