# OpenReview forum: "Learning to Reason via Mixture-of-Thought for Logical Reasoning"
_ICLR.cc/2026/Conference — ICLR 2026 Poster_

### Official Review · Reviewer_Fz8C · 2025-10-30

**Soundness:** 3
**Presentation:** 3
**Contribution:** 3
**Rating:** 6
**Confidence:** 4

**Summary:**

This paper proposes an approach to improve the logical-reasoning capabilities of LLMs by incorporating different reasoning modalities into the decision process. Study aims treat natural language, code and truth table as different reasoning interfaces and apply majority voting to determine model's output to a logical reasoning problem after training the model on multiple reasoning modalities. Performance of the proposed approach has been evaluated on FOLIO, ProofWriter and ProverQA benchmarks, compared with Gemma-2-2B/9B and Qwen-2.5-7B-Instruct.

**Strengths:**

- The study harvests different reasoning interfaces and connects NL reasoning with symbolic reasoning in a single framework.
- Clear performance superiority compared to Qwen-2.5-7B-Instruct, a model capable of reasoning.
- Proposed approach does not require any external teachers.
- Uses interpretable reasoning interfaces, provides space for interpretability studies.

**Weaknesses:**

- Gemma-2-2B/9B models are inherently not proficient in reasoning tasks, their pre-training lacks exposure to used reasoning interfaces (code and truth table).
- Evaluation domain is restricted to logical reasoning which is not the most suitable domain for harvesting different multimodal reasoning interfaces, specifically code.
- Although it is symbolic, truth-table's capability and suitability as a reasoning interface is not convincing.
- Majority voting is a shallow approach for combining the outcomes of reasoning interfaces. It is not suitable for different reasoning problem types.

**Questions:**

- Include comparisons with models better at reasoning.
- Additional experiments on the efficiency of truth-table as a reasoning interface would proof its contribution.
- Evaluation with more comprehensive reasoning benchmarks (e.g. math reasoning) would highlight this approach's capabilities and limitations.

---

> ### Author Response · Authors · 2025-11-23
> **Response to Reviewer Fz8C**
>
> We thank the reviewer for positive and constructive feedback on our work. We have summarized your key concerns and addressed them one by one below.
>
> **Q1: Concerns about Gemma models' reasoning capability and request for stronger models.**
> To validate our method on stronger backbones, we conducted additional experiments using Qwen-3-4B-Instruct-2507, a model with significantly stronger inherent reasoning capabilities than Gemma.
> The results are shown below. We can observe: 1) Even on this strong baseline (83.3%), MoT achieves significant gains, pushing the performance to 89.2% (+5.9%); 2) Notably, our MoT Best Single-Thought (using only 1 inference pass) achieves 87.2%, which significantly outperforms the Baseline + Self-Consistency (3 votes) at 85.2%; and 3)
> MoT with mixture-of-thought inference further boosts performance to 89.2%. Notably, Self-Consistency on the single-thought modality saturates at 87.2% (showing no gain from 87.2% to 87.2% with SC@3). This confirms that the performance boost comes from the diversity of thoughts enabled by different modalities (synergy) rather than just increasing sample quantity. We have similar observations on Proofwriter.
> | Model                         | FOLIO | ProofWriter |
> |-------------------------------|-------|-------------|
> |Reference Score (Only Evaluation)              |  |            |
> |Qwen3-8B (thinking)              | 85.2  | 91.3           |
> |Qwen3-4B (thinking)              | 83.3  | 86.8           |
> |Qwen3-8B (non-thinking)               | 77.3 | 78.3           |
> |Qwen3-4B (non-thinking)               |75.4  | 76.5           |
> |Our new Exp on  Qwen-3-4B-Instruct-2507               |  |           |
> | Qwen-3-4B-Instruct-2507 (nl)                 | 83.3  | 79.7           |
> | Qwen-3-4B-Instruct-2507 (nl) + SC@3          | 85.2  | 82.2           |
> | MoT (best single thought)       | 87.2  | 83.0           |
> | MoT (best single thought) + SC@3| 87.2  | 83.8           |
> | MoT (mixture-of-thought)        | 89.2  | 86.0           |
>
> **Q2: Majority voting is a shallow approach... not suitable for different reasoning problem types.**
>
> We agree that majority voting relies on exact answer matching and is indeed less suitable for open-ended generation or tasks without definitive answers.
> However, we would like to highlight two points regarding the applicability of our framework: 1) In the specific domain of logical reasoning (the focus of this paper), tasks typically have discrete, deterministic outputs (e.g., True/False/Unknown in FOLIO). In this context, majority voting is a standard and effective proxy for reasoning consistency; and 2) Crucially, while the inference aggregation (voting) may be task-dependent, our Self-Evolving Training framework is general. For tasks where voting is inapplicable (e.g., open-ended reasoning), users can simply utilize the MoT Single-Thought model. As demonstrated in our new Qwen-3-4B experiments (Table above ) and original Table 2, the MoT Best Single-Thought can achieve significant performance improvement against corresponding baselines. Furthermore, we are interested in developing more advanced aggregation approaches (e.g., aggregator models) to better leverage different reasoning modalities in future work.
>
> **Q3: Truth-table's capability and efficiency as a reasoning interface.**
>
> We appreciate the reviewer's request for additional clarification on the unique contribution and efficiency of the Truth-Table (TT) modality. We would like to clarify that the current paper already provides detailed analyses demonstrating TT's effectiveness, which we summarize below:
> - **Contribution to Model’s Intrinsic Reasoning Capability**: TT is a core component of our self-evolving training process. Its inclusion significantly boosts the intrinsic reasoning capability of the underlying model, as evidenced by experiments in Table 3.
> - **Efficiency in Inference: TT contributes to the overall MoT system's efficiency**. Our analysis shows TT helps increase the coverage of correct answers during inference. (Section 3.4)
> - **Error analysis: TT provides a crucial structural complement to Natural Language (NL) reasoning**. Our original error analysis and case studies demonstrate that TT systematically enumerates logical cases, helping to overcome key failure modes prevalent in NL reasoning, such as "invalid converse" and "missing branches". (Section 4)

---

> > ### Author Response · Authors · 2025-11-23
> > **Response to Reviewer Fz8C (Continue)**
> >
> > **Q4: Domain restriction (Logic) & Generalization to other benchmarks (e.g., Math).**
> >
> > We acknowledge that our primary focus is on Logical Reasoning, where interfaces like Truth Tables and Code are most naturally aligned. To address the concern about generalization and discuss the capabilities and limitations of our approach:
> > - **Safety on General Reasoning**: We evaluated our MoT-trained model on the GSM8K (Math Reasoning) and StrategyQA benchmarks using the Natural Language (NL) modality. The model achieves comparable or better performance to the baseline. This indicates that MoT is "safe": while it specializes the model for logic tasks (via TT/Code), it does not overfit to logical reasoning or harm the model's general reasoning capabilities in other domains.
> > - **Discussion on Limitations**: We agree that the Truth-Table modality is domain-specific and less applicable to open-ended math problems. However, this highlights a key advantage of our framework's modality flexibility: for domains where symbolic interfaces (like TT) are not suitable, the model can seamlessly fall back to the NL modality without significant performance degradation.
> >
> >
> > | Model                   | Setting               | StrategyQA Acc | GSM8K Acc | FOLIO Acc | ProofWriter Acc |
> > |-------------------------|------------------------|----------------|-----------|-----------|------------------|
> > | **Gemma-2-2B-Instruct** | Baseline               | 81.7           | 52.1      | 42.4      | 39.8             |
> > |                         | MoT (Single-Thought)   | 81.2           | 51.5      | 61.1      | 62.7             |
> > |                         | Δ (MoT - Base)         | -0.5           | -0.6      | +18.7     | +22.9            |
> > |-------------------------|------------------------|----------------|-----------|-----------|------------------|
> > | **Gemma-2-9B-Instruct** | Baseline               | 93.2           | 85.2      | 69.5      | 61.2             |
> > |                         | MoT (Single-Thought)   | 92.3           | 84.2      | 76.9      | 69.5             |
> > |                         | Δ (MoT - Base)         | -0.9           | -1.0      | +7.4      | +8.3             |
> > |-------------------------|------------------------|----------------|-----------|-----------|------------------|
> > | **Qwen-2.5-7B-Instruct**| Baseline               | 88.1           | 87.6      | 71.9      | 60.5             |
> > |                         | MoT (Single-Thought)   | 89.5           | 86.9      | 75.9      | 69.2             |
> > |                         | Δ (MoT - Base)         | +1.4           | -0.7      | +4.0      | +8.7             |

---

### Official Review · Reviewer_yjSj · 2025-10-31

**Soundness:** 3
**Presentation:** 4
**Contribution:** 3
**Rating:** 4
**Confidence:** 4

**Summary:**

This paper proposes a method to fine-tune LLMs to reason in three different “thought modalities” for solving logical reasoning problems. They train in a self-evolving manner where they generate thoughts and use thoughts which resulted in correct answers as the data for the next iteration. To perform inference, the model responds in each of the three modalities and then takes the majority answer as the final answer. They show that this approach improves in accuracy over using any one of the thought modalities as well as helps on the hardest of reasoning problems.

**Strengths:**

* The “thought modality” fine-tuning appears to be novel. The exploration of fine-tuning on different “thought modalities” such as coding and CoT has been extensively studied in terms of post-training on different datasets (coding problems, math word problems, etc.), but I am not aware of work which fine tunes on different solution methods for the same dataset.
* The paper is well written and easy to follow.
* The problem of improving reasoning performance by increasing “thought” diversity is important and this approach to fine-tuning on different solution strategies can be impactful.

**Weaknesses:**

* Experiments are only performed on two datasets and results are missing standard deviations.
* The natural language and code reasoning modalities seem highly general, but the truth table modality seems specific to the two benchmarks used in this paper. If using MoT on a different reasoning dataset, this could potentially harm performance.
* The experiments are a little hard to evaluate due to the confounder of the number of samples which different methods use and the different models used. For instance in Table 2, the last row MoT method takes the majority answer from 3 samples, but I assume that the line above that (MoT Single-Thought) only uses one sample. Also, Logic-LM is only evaluated with GPT-4.

**Questions:**

1. Does MoT Single-Thought use majority voting from three samples? If not, I think this is an important baseline as well.
2. How should one come up with the relevant reasoning modalities? Are there a set of general reasoning modalities that could work for any reasoning problem?

---

> ### Author Response · Authors · 2025-11-23
> **Response to Reviewer yjSj**
>
> We sincerely thank Reviewer yjSj for the thoughtful and constructive feedback. They are invaluable for further improving the quality of our paper. We hope that you might consider raising your score in light of these clarifications.
>
> > Experiments are only performed on two datasets and results are missing standard deviations.
>
> Thank you for your suggestions. To directly address these concerns, we have taken two actions:
> - **Standard Deviation**: Following prior work in LLM reasoning (e.g., LINC, Logic-LM, LoT), Table 2 in the main paper reports performance under the standard single-run inference setting, as cost of experiments on LLMs are very high. To solve your concern, we additionally conduct a robustness analysis under three inference seeds (42, 123, 2025) and show the results below. The variance is small, and all trends remain fully consistent with our Table 2, confirming the stability of our conclusions. We will add this additional analysis in the Appendix.
> - **Broader Evaluation**: We have expanded the evaluation to include the GSM8K (Mathematical Reasoning) and StrategyQA (combination of Commonsense, Logical and Mathematical Reasoning) benchmarks (as shown in Table R2). For these two generalization tasks, we use natural language modality only for inference as code and truth tables may not be suitable for these two tasks. From Table R2, we can see that the MoT framework can significantly improve logical reasoning ability while  maintaining general reasoning ability. This also highlights the modality flexibility of our MoT framework: the user can select the most appropriate reasoning modality during inference. proving that the MoT training is safe and does not restrict the model's application to logic-only domains.
> We provided the results below.
> ### Table R1:  Additional robust analysis on the FOLIO and ProofWriter benchmarks under three seeds (2025, 123, 42)
> | Model | Method Type | Reasoning Modality | FOLIO | ProofWriter | Avg |
> |-------|-------------|---------------------|--------|--------------|------|
> | **(A) Prior SOTA Approach** |  |  |  |  |  |
> | GPT-4 | Logic-LM | - | 78.9 | 79.7 | 79.3 |
> | GPT-4 | CoT (Vanilla) | - | 70.6 | 68.1 | 69.4 |
> | **(B) Base Model: Gemma-2-2B-It** |  |  |  |  |  |
> | Gemma-2-2B-It (3-Shot) | Single-Thought | Best (nl) | 41.9 ± 1.1 | 39.4 ± 1.4 | 40.7 |
> | Gemma-2-2B-It @ 3 (3-Shot) | Single-Thought | Best (nl) | 42.7 ± 2.5 | 38.8 ± 1.1 | 40.8 |
> | **MoT (0-Shot)** | Single-Thought | Best | 60.6 ± 1.8 | 61.4 ± 1.8 | 61.0 |
> | **MoT (0-Shot)** | Mixture-of-Thought | All | **61.8 ± 0.6** | **63.9 ± 0.9** | **62.9** |
> | **(C) Base Model: Gemma-2-9B-It** |  |  |  |  |  |
> | Gemma-2-9B-It (3-Shot) | Single-Thought | Best (nl) | 70.5 ± 1.1 | 60.1 ± 0.3 | 65.3 |
> | Gemma-2-9B-It @ 3 (3-Shot) | Single-Thought | Best (nl) | 73.4 ± 0.7 | 62.5 ± 0.2 | 68.0 |
> | **MoT (0-Shot)** | Single-Thought | Best | 77.7 ± 0.6 | 68.4 ± 0.8 | 73.1 |
> | **MoT (0-Shot)** | Mixture-of-Thought | All | **79.3 ± 0.4** | **71.0 ± 0.5** | **75.2** |
> | **(D) Base Model: Qwen2.5-7B-Instruct** |  |  |  |  |  |
> | Qwen2.5-7B-Instruct (3-Shot) | Single-Thought | Best (nl) | 72.4 ± 1.9 | 60.8 ± 1.9 | 66.6 |
> | Qwen2.5-7B-Instruct @ 3 (3-Shot) | Single-Thought | Best (nl) | 73.1 ± 0.5 | 65.2 ± 0.8 | 69.2 |
> | **MoT (0-Shot)** | Single-Thought | Best | 76.5 ± 0.8 | 70.5 ± 1.0 | 73.5 |
> | **MoT (0-Shot)** | Mixture-of-Thought | All | **79.0 ± 0.6** | **72.2 ± 0.3** | **75.6** |
>
>
>
> ### Table R2: More evaluations on GSM8K and StrategyQA
> | Model                 | Setting    | StrategyQA Accuracy| GSM8K Accuracy | FOLIO Accuracy | ProofWriter Accuracy |
> |-----------------------|------------|----------------|----------------|----------------|----------------|
> | **Gemma-2-2B-Instruct** | Baseline   | 81.7 | 52.1          | 42.4 |  39.8 |
> |                       | MoT  (Single-Thought)  | 81.2       | 51.5          | 61.1  | 62.7  |
> |                       | Δ (MoT - Base) | -0.5 | -0.6      | + 18.7 | + 22.9 |
> |-----------------------|------------|----------------|----------------|----------------|----------------|
> | **Gemma-2-9B-Instruct** | Baseline   | 93.2 | 85.2          | 69.5 | 61.2 |
> |                       | MoT (Single-Thought)  | 92.3     | 84.2        |  76.9 | 69.5 |
> |                       | Δ (MoT - Base) | -0.9 | -1.0      | + 7.4 | + 8.3 |
> |-----------------------|------------|----------------|----------------|----------------|----------------|
> | **Qwen-2.5-7B-Instruct** | Baseline   | 88.1 | 87.6          |  71.9 | 60.5 |
> |                       | MoT   (Single-Thought) |  89.5      | 86.9          | 75.9 | 69.2 |
> |                       | Δ (MoT - Base)  | +1.4 |  -0.7     | + 4.0 | + 8.7 |

---

> > ### Author Response · Authors · 2025-11-23
> > **Response to Reviewer yjSj (continue.)**
> >
> > > The natural language and code reasoning modalities seem highly general, but the truth table modality seems specific to the two benchmarks used in this paper. If using MoT on a different reasoning dataset, this could potentially harm performance.
> >
> > Thank you for raising this concern. We acknowledge truth table modality is specific to logical reasoning which is the focus of our paper. However, our MoT should be viewed as a general reasoning framework that integrates three modalities, e.g., natural language, code, and symbolic structures, during training and inference. Truth tables are an instance of the symbolic interface for logical reasoning, not a requirement of MoT. To flexibly enable MoT trained models to work on other domains, the user can select the most appropriate reasoning modality during inference for specific tasks, e.g., nl modality for gsm8k and StrategyQA. Our results (Table R2) also show that MoT models with NL modality can achieve better or comparable results compared to baselines. Therefore, using MoT on a different reasoning dataset  could not harm performance.
> >
> > > The experiments are a little hard to evaluate due to the confounder of the number of samples which different methods use and the different models used. For instance in Table 2, the last row MoT method takes the majority answer from 3 samples, but I assume that the line above that (MoT Single-Thought) only uses one sample. Also, Logic-LM is only evaluated with GPT-4. & Does MoT Single-Thought use majority voting from three samples? If not, I think this is an important baseline as well.
> >
> > Thanks for raising this concern. We clarify the intended design and provide additional experiments to remove any potential confounds regarding sample count.
> >
> > **Upper Bound Reference (GPT-4 Logic-LM)**: This row serves purely as a reference upper bound to show that MoT enables open-source 2B–9B models to approach the performance of a GPT-4-based system.
> >
> > **Isolating Training Effect**: MoT Single-Thought uses one sample, identical to standard CoT, to isolate the training effect of MoT under a matched inference setup. Furthermore, MoT with MoT inference (3 sample) compares baseline with SC@3 to show persistent effectiveness under budget constraints of three samples.
> >
> > To further highlight our mixture-of-thought inference, we additionally include MoT Single-Thought + SC@3. The results are below. We will add the results in Table 2 in the revised version of the paper.
> >
> >
> > ### Base Model: Gemma-2-2B-It
> > | Model | Method Type | Reasoning Modality | FOLIO | ProofWriter | Avg |
> > |-------|-------------|---------------------|--------|--------------|------|
> > | Gemma-2-2B-It (3-Shot) | Single-Thought | Best (nl) | 42.4 | 39.8 | 41.1 |
> > | Gemma-2-2B-It @ 3 (3-Shot) | Single-Thought | Best (nl) | 45.3 | 38.8 | 42.1 |
> > | MoT (0-Shot) | Single-Thought | Best | 61.1 |  62.7 | 61.9 |
> > | MoT (0-Shot) SC@3 | Single-Thought | Best |62.1 | 60.8 | 61.5 |
> > | MoT (0-Shot) | Mixture-of-Thought | All | 62.6 | 65.0 | 63.8 |
> >
> >
> > ### Base Model: Gemma-2-9B-It
> > | Model | Method Type | Reasoning Modality | FOLOLIO | ProofWriter | Avg |
> > |-------|-------------|---------------------|--------|--------------|------|
> > | Gemma-2-9B-It (3-Shot) | Single-Thought | Best (nl) | 69.5 | 61.2 | 65.4 |
> > | Gemma-2-9B-It @ 3 (3-Shot) | Single-Thought | Best (nl) | 72.9 | 62.7 | 67.8 |
> > | MoT (0-Shot) | Single-Thought | Best | 76.9 | 69.5 | 73.2 |
> > | MoT (0-Shot) SC@3 | Single-Thought | Best | 75.4 | 70.5 | 73.0 |
> > | MoT (0-Shot) | Mixture-of-Thought | All | 78.9 | 70.7 | 74.8 |
> >
> >
> > ### Base Model: Qwen2.5-7B-Instruct
> > | Model | Method Type | Reasoning Modality | FOLIO | ProofWriter | Avg |
> > |-------|-------------|---------------------|--------|--------------|------|
> > | Qwen2.5-7B-Instruct (3-Shot) | Single-Thought | Best (nl) | 71.9 | 60.5 | 66.2 |
> > | Qwen2.5-7B-Instruct @ 3 (3-Shot) | Single-Thought | Best (nl) | 73.4 | 65.8 | 69.6 |
> > | MoT (0-Shot) | Single-Thought | Best | 75.9 | 69.2 | 72.6 |
> > | MoT (0-Shot) SC@3 | Single-Thought | Best | 77.3 | 70.5 |  73.9 |
> > | MoT (0-Shot) | Mixture-of-Thought | All | 78.3 | 71.8 | 75.1 |
> >
> >
> > > How should one come up with the relevant reasoning modalities? Are there a set of general reasoning modalities that could work for any reasoning problem?
> >
> > MoT should be viewed as a general framework rather than a fixed set of modalities. In practice, most reasoning tasks can be naturally covered by three broad categories: 1) Natural language (free-form explanation) 2) Code reasoning (algorithmic steps) and Symbolic structures (logic tables, algebraic derivations, equation chains, state transitions, etc.)
> > The symbolic category is intentionally domain-dependent: different domains naturally instantiate different symbolic formats. For logical reasoning this takes the form of truth tables, while in mathematical reasoning it may correspond to algebraic or equation-based derivations.

---

### Official Review · Reviewer_1P9u · 2025-11-01

**Soundness:** 3
**Presentation:** 3
**Contribution:** 2
**Rating:** 4
**Confidence:** 4

**Summary:**

This paper proposes a framework called Mixture-of-Thought (MoT), which aims to enable an LLM to jointly learn and utilize three complementary reasoning modalities (natural language, code, and a novel truth-table modality) within a unified model via a self-evolving training process. This method significantly outperforms single-modality baselines on standard logical reasoning benchmarks (FOLIO, ProofWriter), particularly by introducing the truth-table modality to address key bottlenecks in natural language reasoning, such as "missing branches" and "invalid converse".

**Strengths:**

- The paper's motivation is clear and well-supported. The fusion of three modalities is experimentally shown to effectively enhance model performance. Notably, the design of the truth-table modality is driven by a thorough error analysis shown in Figure 1c, which pointedly addresses key bottlenecks of NL CoT.

- The paper proposes a self-evolving training method that enables a single LLM to jointly learn and synergistically utilize multiple reasoning modalities, addressing the "modality-blind" problem of existing methods during the training phase.

**Weaknesses:**

- The generalization ability of the truth-table (TT) modality is questionable. This modality is currently highly customized for logical reasoning and has only been validated on two logical reasoning benchmarks. I suggest the authors discuss or explore the framework's potential application in other reasoning domains, such as mathematical or commonsense reasoning.

- Experiments in Table 3, Figure 3b, Figure 4, and Figure 7 show that the code modality almost always performs worse when used alone. However, Table 3 also shows that including it in joint training still yields performance gains. It will be better to provide supplementary qualitative or quantitative analysis on: (i) What are the specific advantages of the code modality? (ii) What specific problems or sample types does it solve that the other two modalities cannot?

- It can benefit from further analyzing the application scenarios for each modality. If it's possible to identify which problems are best suited for a specific modality, perhaps a dynamic routing strategy could be employed at inference time instead of performing three separate inferences for all problems (as in MoT-Inference), thereby optimizing the high inference cost.

- Please provide a more detailed explanation of the "reason to prune" step within the Truth-Table CoT, either in the main paper or the appendix. The current description, "eliminates rows that violate any premise through reasoning via LLMs", is too abstract and lacks sufficient implementation details for reproducibility.

- Including a post-MoT-training error analysis is necessary, similar to the pre-training analysis in Figure 1c. This would help verify whether MoT training not only improves ensemble performance but also enhances the NL modality itself, for example, by reducing the frequency of "missing branch" and "invalid converse" errors.

Some recent relevant works should be included to discuss or compare with the proposed method:

[1] Tree of Thoughts: Deliberate Problem Solving with Large Language Models

[2] Cumulative Reasoning with Large Language Models

[3] DetermLR: Augmenting LLM-based Logical Reasoning from Indeterminacy to Determinacy

**Questions:**

See above.

---

> ### Author Response · Authors · 2025-11-23
> **Response to Reviewer 1P9u**
>
> We sincerely thank the reviewer for the constructive feedback on our paper. We have addressed your concerns one by one below.
>
> >The generalization ability of the truth-table (TT) modality is questionable. This modality is currently highly customized for logical reasoning and has only been validated on two logical reasoning benchmarks. I suggest the authors discuss or explore the framework's potential application in other reasoning domains, such as mathematical or commonsense reasoning
>
> We thank the reviewer for raising this important point. We would like to clarify that MoT is a general mixture-of-thought reasoning framework, integrating code, natural language and symbolic modalities. As our main focus in this paper is to improve logical reasoning ability of LLMs, we specifically design truth-table modality as a domain-specific instantiation of the symbolic interface because such a modality is complementary to the natural language modality from analysis in Figure 1.
> Now we discuss the framework's potential application in other reasoning domains from two aspects:
> - **Our MoT model trained on logic maintains its general reasoning ability with the natural language modality while significantly improving logical reasoning with the Code and truth-table modality.** We present the results on GSM8K and StrategyQA below, which are widely-adopted benchmarks for studying mathematical reasoning and commonsense reasoning respectively. Notably, we use the natural language modality alone to solve these two tasks.
> - We can directly enable MoT to work for other reasoning domains by designing specific instantiation of the symbolic modality, e.g., equation derivations for math reasoning tasks.
> We have included this discussion in the revised version of our paper.
>
>
> | Model                 | Setting    | StrategyQA Accuracy| GSM8K Accuracy | FOLIO Accuracy | ProofWriter Accuracy |
> |-----------------------|------------|----------------|----------------|----------------|----------------|
> | **Gemma-2-2B-Instruct** | Baseline   | 81.7 | 52.1          | 42.4 |  39.8 |
> |                       | MoT  (Single-Thought)  | 81.2       | 51.5          | 61.1  | 62.7  |
> |                       | Δ (MoT - Base) | -0.5 | -0.6      | + 18.7 | + 22.9 |
> |-----------------------|------------|----------------|----------------|----------------|----------------|
> | **Gemma-2-9B-Instruct** | Baseline   | 93.2 | 85.2          | 69.5 | 61.2 |
> |                       | MoT (Single-Thought)  | 92.3     | 84.2        |  76.9 | 69.5 |
> |                       | Δ (MoT - Base) | -0.9 | -1.0      | + 7.4 | + 8.3 |
> |-----------------------|------------|----------------|----------------|----------------|----------------|
> | **Qwen-2.5-7B-Instruct** | Baseline   | 88.1 | 87.6          |  71.9 | 60.5 |
> |                       | MoT   (Single-Thought) |  89.5      | 86.9          | 75.9 | 69.2 |
> |                       | Δ (MoT - Base)  | +1.4 |  -0.7     | + 4.0 | + 8.7 |
>
> > Experiments in Table 3, Figure 3b, Figure 4, and Figure 7 show that the code modality almost always performs worse when used alone. However, Table 3 also shows that including it in joint training still yields performance gains. It will be better to provide supplementary qualitative or quantitative analysis on: (i) What are the specific advantages of the code modality? (ii) What specific problems or sample types does it solve that the other two modalities cannot?
>
> Thanks for your suggestions. To directly answer your question about the function of code modality, we analyze the case that is solely solved by the code modality as shown in Table 8 (Appendix). We find the Code Modality is superior to the other two where high reasoning complexity is required. These examples are characterized by lots of premises and variables and require long inference chains. In this scenario,  Natural Language modality suffers from error accumulation or hallucination, while the Truth-Table modality faces challenges in building truth tables. The Code Modality fills this critical gap by providing a scalable and formally expressive encoding. However, we also acknowledge that for complex problems that cannot be easily transformed into code structures, the Code Modality still falls short. This also explains why code modality almost always performs worse when used alone.

---

> > ### Author Response · Authors · 2025-11-23
> > **Response to Reviewer 1P9u (continue.)**
> >
> > > It can benefit from further analyzing the application scenarios for each modality. If it's possible to identify which problems are best suited for a specific modality, perhaps a dynamic routing strategy could be employed at inference time instead of performing three separate inferences for all problems (as in MoT-Inference), thereby optimizing the high inference cost.
> >
> > We thank the reviewer for this insightful suggestion. We agree that employing a dynamic routing strategy is crucial for optimizing the high inference cost of MoT. Our analysis confirms the complementarity of the modalities: the Truth Table (TT) modality excels at specific formal logical structures (like Modus Tollens and disjunctions), while the Code Modality is uniquely superior in handling high-complexity scenarios where TT faces difficulty in building accurate truth table and NL suffers from error accumulation or hallucination. This clear distinction demonstrates that designing a problem-specific routing strategy based on logical structure and problem complexity is highly feasible and will be a valuable direction for our future work. Notably, HybridMind [1] attempted to train such a router, although direct-prompting or fine-tuning with distilled data does not make the routing accuracy very high.
> >
> > [1] HYBRIDMIND: Meta Selection of Natural Language and Symbolic Language for Enhanced LLM Reasoning.
> >
> > > Please provide a more detailed explanation of the "reason to prune" step within the Truth-Table CoT, either in the main paper or the appendix. The current description, "eliminates rows that violate any premise through reasoning via LLMs", is too abstract and lacks sufficient implementation details for reproducibility.
> >
> > We sincerely thank the reviewer for this crucial feedback. We agree that clear implementation details are vital for the reproducibility and transparency of our method. We have comprehensively clarified the 'Reason to Prune' functionality and have included its complete, detailed implementation in Appendix.
> >
> > We leverage Few-Shot Prompting to guide the LLM to perform Constraint-Aware Deductive Reasoning. The specific mechanism for this implicit pruning is as follows:
> > - Structured Prompting: The provided few-shot examples enforce a strict structure (Logical Chain and Truth Table tags), compelling the LLM to follow this goal-oriented reasoning pattern.
> > - Logical Chain Induction: The LLM is prompted to perform multi-step deduction (e.g., $T \Rightarrow H \Rightarrow A \Rightarrow B$) based on the premises' transitive properties. This process itself is a form of Logical Grounding, where each derived truth value (e.g., $H=\text{True}$) must be consistent with all preceding assignments and premises.
> > - Implicit Pruning: The deduction is inherently constraint-aware. The LLM only derives and outputs a set of variable assignments that are logically consistent with the entire premise set. This ensures that the derived assignments (the final Truth Table row) represent the unique (or minimal) logically plausible world.
> > - Efficient Output: By performing this deduction, the LLM directly generates the required minimal truth table, thereby implicitly 'pruning' all logically inconsistent rows from the potential $2^n$ search space.
> >
> > This design integrates the LLM's powerful reasoning capability with the rigor of formal logic, resulting in a reasoning path that is more word-efficient and possesses higher logical verifiability than the traditional free-form CoT.
> >
> > > Including a post-MoT-training error analysis is necessary, similar to the pre-training analysis in Figure 1c. This would help verify whether MoT training not only improves ensemble performance but also enhances the NL modality itself, for example, by reducing the frequency of "missing branch" and "invalid converse" errors.
> >
> > We thank the reviewer for the suggestion. We have added post-MoT-training error analysis.  It shows that all error categories decrease after MoT training, with the most significant reductions appearing in commonsense injection and factual misquote error types. This indicates that MoT enhances the logical consistency of the NL reasoning modality. However, the remaining missing branch (MB) and  invalid-converse (IC) errors are still the most structurally challenging cases. NL alone cannot completely deal with them. This observation highlights the necessity of MoT-Inference. The inference-time collaboration among NL, Code, and TT modalities can enable the model to further reduce these difficult error types, compensating for NL’s limitations.
> > | Model                    | IC   | MB   | CI   | FC   |
> > | ------------------------ | ---- | ---- | ---- | ---- |
> > | Qwen-2.5-7B-Instruct     | 22   | 18   | 12   | 12   |
> > | Qwen-2.5-7B-Instruct-Mot | 21   | 16   | 7    | 8    |

---

> > > ### Author Response · Authors · 2025-11-23
> > > **Response to Reviewer 1P9u (continue.)**
> > >
> > > > Some recent relevant works should be included to discuss or compare with the proposed method:
> > > [1] Tree of Thoughts: Deliberate Problem Solving with Large Language Models
> > > [2] Cumulative Reasoning with Large Language Models
> > > [3] DetermLR: Augmenting LLM-based Logical Reasoning from Indeterminacy to Determinacy
> > >
> > > Thank you for pointing out these related works. We have added them into the revised Related Work section and provided conceptual comparisons.
> > >
> > > Tree-of-Thought (ToT) improves reasoning through search over multiple paths in a tree structure. Cumulative Reasoning (CR) accumulates verified propositions in a DAG through a Proposer–Verifier–Reporter scheme. DetermLR enhances reasoning through premise identification, prioritization, and memorization to improve determinacy.
> > >
> > > Our MoT framework addresses a different dimension of the reasoning problem. Instead of modifying the search structure or premise organization, MoT introduces multiple heterogeneous reasoning modalities (NL, code, truth-table) and enables the model to collaboratively learn to reason and reason across them, providing complementary inductive biases that single-modality methods do not capture. Modal diversity is orthogonal to tree search (ToT), cumulative verification (CR), and determinacy modeling (DetermLR).
> > >
> > > We also see some potential opportunities by combining these paradigms. Since ToT/CR/DetermLR operate on tree/graph-based reasoning structures, incorporating modality diversity into such frameworks may be a promising future direction (e.g., routing different modalities for different sub-problem types).
> > >
> > > We thank the reviewer for highlighting these works and believe the added discussion significantly improves the clarity of our positioning.

---

### Official Review · Reviewer_6rwT · 2025-11-02

**Soundness:** 3
**Presentation:** 3
**Contribution:** 3
**Rating:** 6
**Confidence:** 3

**Summary:**

The paper introduces Mixture-of-Thought, a framework that enables large language models to reason using three complementary modalities ( i.e., natural language, code, and a symbolic truth-table format ). Unlike prior methods that train with a single modality and only vary modalities at inference, MoT trains and infers across modalities to exploit their synergy. A self-evolving training phase jointly learns from filtered, self-generated rationales in all three formats, while the inference phase integrates them to produce stronger predictions. The truth-table modality systematically enumerates logical cases, helping mitigate common failure modes of natural-language reasoning. Experiments on logical reasoning benchmarks such as FOLIO and ProofWriter show MoT significantly outperforms strong single-modality chain-of-thought baselines, with average gains up to 11.7 percentage points.

**Strengths:**

1. The paper is well-written and easy to follow.
2. The proposed Self-Evolving MoT Training method is reasonable and effective.
3. Extensive experiments and ablations were conducted on various models and showed strong performance.

**Weaknesses:**

1. The proposed method appears applicable only to specific logical reasoning tasks, and its generalization to broader reasoning tasks or other domains remains uncertain.

**Questions:**

1. To better highlight the impact of MoT inference, I suggest also reporting the Self-Consistency with three votes metric for the MoT with Single-Thought inference results in Table 2.

---

> ### Author Response · Authors · 2025-11-23
> **Response to Reviewer 6rwT**
>
> We thank the reviewer for raising these important points. We agree that a robust evaluation of generalization and a clearer breakdown of the inference gain are essential. We have addressed both concerns via following new experiments and clarifications.
>
> **Regarding Generalization to Broader Reasoning Tasks**: We acknowledge that the core motivation of our work is to significantly improve logical reasoning ability, and the primary datasets (FOLIO, ProofWriter) reflect this focus. However, we have taken two steps to robustly address the concern regarding MoT's general applicability:
> - **MoT training is "safe" and does not impair the model's performance on non-logical domains.** We have conducted additional evaluation on GSM8K (Mathematical Reasoning) and StrategyQA benchmarks (as shown in Table 1). For these tasks, we used only the Natural Language (NL) modality during inference, as the code and truth table modalities may be not very suitable for these domains. We can see the MoT framework can maintain general reasoning ability while significantly improving our targeted logical reasoning tasks. This also highlights the flexibility of our MoT framework. We enable models to learn to reason via multiple reasoning modalities and we can choose suitable reasoning modality to deal with specific tasks.
> - **The MoT framework is conceptually principled with broad applicability**: We emphasize that MoT is designed as a general reasoning framework to integrate multiple modalities—Natural Language (NL), Code Reasoning, and Symbolic Structures—during training and inference. While truth tables are the natural symbolic modality for logical reasoning (our paper's focus), the symbolic category is designed to be domain-dependent. For tasks like mathematical reasoning, the symbolic structure would be instantiated as algebraic derivations or equation chains. With these modified designs, we can directly extend our MoT to improve upon NL reasoning for non-logic reasoning tasks.
>
> Table 1: More evaluations on GSM8K and StrategyQA
> | Model                 | Setting    | StrategyQA Accuracy| GSM8K Accuracy | FOLIO Accuracy | ProofWriter Accuracy |
> |-----------------------|------------|----------------|----------------|----------------|----------------|
> | **Gemma-2-2B-Instruct** | Baseline   | 81.7 | 52.1          | 42.4 |  39.8 |
> |                       | MoT  (Single-Thought)  | 81.2       | 51.5          | 61.1  | 62.7  |
> |                       | Δ (MoT - Base) | -0.5 | -0.6      | + 18.7 | + 22.9 |
> |-----------------------|------------|----------------|----------------|----------------|----------------|
> | **Gemma-2-9B-Instruct** | Baseline   | 93.2 | 85.2          | 69.5 | 61.2 |
> |                       | MoT (Single-Thought)  | 92.3     | 84.2        |  76.9 | 69.5 |
> |                       | Δ (MoT - Base) | -0.9 | -1.0      | + 7.4 | + 8.3 |
> |-----------------------|------------|----------------|----------------|----------------|----------------|
> | **Qwen-2.5-7B-Instruct** | Baseline   | 88.1 | 87.6          |  71.9 | 60.5 |
> |                       | MoT   (Single-Thought) |  89.5      | 86.9          | 75.9 | 69.2 |
> |                       | Δ (MoT - Base)  | +1.4 |  -0.7     | + 4.0 | + 8.7 |
>
>
>
> **Regarding MoT with Single-Thought Self-Consistency baseline**:
> Thank you for the suggestion. We have added the baseline and the results are below.
>
> | Model | Method Type | Reasoning Modality | FOLIO | ProofWriter | Avg |
> |-------|-------------|---------------------|--------|--------------|------|
> | Gemma-2-2B-It (3-Shot) | Single-Thought | Best (nl) | 42.4 | 39.8 | 41.1 |
> | Gemma-2-2B-It @ 3 (3-Shot) | Single-Thought | Best (nl) | 45.3 | 38.8 | 42.1 |
> | MoT (0-Shot) | Single-Thought | Best | 61.1 |  62.7 | 61.9 |
> | MoT (0-Shot) SC@3 | Single-Thought | Best |62.1 | 60.8 | 61.5 |
> | MoT (0-Shot) | Mixture-of-Thought | All | 62.6 | 65.0 | 63.8 |
>
>
> | Model | Method Type | Reasoning Modality | FOLOLIO | ProofWriter | Avg |
> |-------|-------------|---------------------|--------|--------------|------|
> | Gemma-2-9B-It (3-Shot) | Single-Thought | Best (nl) | 69.5 | 61.2 | 65.4 |
> | Gemma-2-9B-It @ 3 (3-Shot) | Single-Thought | Best (nl) | 72.9 | 62.7 | 67.8 |
> | MoT (0-Shot) | Single-Thought | Best | 76.9 | 69.5 | 73.2 |
> | MoT (0-Shot) SC@3 | Single-Thought | Best | 75.4 | 70.5 | 73.0 |
> | MoT (0-Shot) | Mixture-of-Thought | All | 78.9 | 70.7 | 74.8 |
>
>
> | Model | Method Type | Reasoning Modality | FOLIO | ProofWriter | Avg |
> |-------|-------------|---------------------|--------|--------------|------|
> | Qwen2.5-7B-Instruct (3-Shot) | Single-Thought | Best (nl) | 71.9 | 60.5 | 66.2 |
> | Qwen2.5-7B-Instruct @ 3 (3-Shot) | Single-Thought | Best (nl) | 73.4 | 65.8 | 69.6 |
> | MoT (0-Shot) | Single-Thought | Best | 75.9 | 69.2 | 72.6 |
> | MoT (0-Shot) SC@3 | Single-Thought | Best | 77.3 | 70.5 |  73.9 |
> | MoT (0-Shot) | Mixture-of-Thought | All | 78.3 | 71.8 | 75.1 |

---

### Author Response · Authors · 2025-12-01
**Summary and Clarification of Revision and Rebuttal process.**

Dear reviewers and ACs,

Thank you for your constructive feedback. We appreciate that reviewers consistently highlighted several core strengths of the paper:

- the clear and well-supported motivation, with the truth-table modality grounded in the NL-CoT error analysis in Figure 1c (1P9u);

- the conceptual contribution of Self-Evolving MoT Training, enabling a single model to jointly learn and synergistically utilize NL, code, and symbolic reasoning (6rwT, 1P9u, yjSj, Fz8C); and

- strong empirical results with extensive ablations across backbones, consistently outperforming strong baselines such as Qwen-2.5-7B-Instruct (6rwT, Fz8C).

During the rebuttal period, we provided detailed clarifications and conducted substantial additional analyses to address raised concerns:

- **Stronger backbone evaluation**: Added MoT experiments on Qwen-3-4B-Instruct. (Table 2)

- **Variance and robustness**: Added three-seed analysis (42/123/2025) with standard deviations. (Appendix E.1; Table 2)

- **Generalization beyond logic**: Added StrategyQA and GSM8K experiments and expanded discussion on MoT’s applicability to broader reasoning tasks. (Sec. 4.2; Table 4)

- **Completeness of related work**: Added missing comparisons with ToT, Cumulative Reasoning, and DetermLR. (Sec. 5)

- **Role of the code modality**: Added further qualitative and quantitative analysis illustrating code-only strengths on high-complexity instances. (Sec. 4.3)

- **Post-training NL error analysis**: Added the post-MoT breakdown mirroring Figure 1c to show reductions across major error categories. (Appendix E.2; Table 6)

- **Inference fairness**: Added MoT Single-Thought + SC@3 baseline to isolate MoT-Inference effects. (Sec. 3.2; Table 2)

- **Truth-table CoT reproducibility**: Added detailed explanation of the “reason-to-prune” mechanism in TT-CoT. (Appendix D.4)

While reviewers did not provide further follow-up comments, likely due to time and system constraints, we ensured that all concerns were thoroughly addressed through new experiments, detailed analyses, and revisions to the manuscript.

Thank you again to all reviewers, ACs, and PCs for your time, feedback, and engagement throughout the process.

---

### Meta-Review · Area_Chair_4X5U · 2026-01-10

**Summary:**

The paper introduces Mixture-of-Thought (MoT), a framework allowing a single LLM to jointly learn and reason across natural language, code, and symbolic truth-tables for logical reasoning. Reviewers commented positively on the clear motivation, solid design, and strong empirical results, noting significant gains over baselines on FOLIO and ProofWriter. The main concerns centered on (1) the generalization of the truth-table modality beyond logical reasoning, (2) the contribution of the code modality, (3) clarity and reproducibility of the truth-table pruning mechanism, and (4) fair isolation of MoT inference gains. The authors addressed these points in the revision. They added evaluations on GSM8K and StrategyQA showing that MoT training preserves general reasoning ability, clarified that the truth-table is a domain-specific instantiation of a broader symbolic interface, and provided additional analyses demonstrating that the code modality contributes complementary strengths on high-complexity cases. They also added MoT Single-Thought + self-consistency baselines, expanded methodology, post-training error analysis, stronger backbone experiments, variance analysis, and missing related work comparisons.

**Reviewer Concerns:**

Most reviewer concerns were addressed in the rebuttal and revision. The authors added GSM8K and StrategyQA evaluations and clarified that the truth-table is a domain-specific instantiation of a general symbolic interface, addressing generalization concerns. They provided additional analyses to clarify the role of the code modality, added MoT Single-Thought + self-consistency baselines to isolate inference gains, expanded methodological details for reproducibility, and included post-training error analysis, stronger backbones, robustness experiments, and missing related work.

However, some concerns remain unaddressed or only partially addressed. MoT has not yet been empirically demonstrated with alternative symbolic modalities beyond truth tables, and inference-time aggregation is still based on simple majority voting rather than learned or adaptive routing.
Reviewer Scores

**Reviewer Scores:**

Reviewer 6rwT (initial 6): Likely unchanged score (6).
Reviewer 1P9u (4): Likely unchanged score (4).
Reviewer yjSj (4): Likely unchanged score (4).
Reviewer Fz8C (6): Likely unchanged score (6).

---

### Decision · Program_Chairs · 2026-01-26

Accept (Poster)